

**Australian tidal currents – assessment of a barotropic model (COMPAS v1.3.0 rev6631) with an unstructured grid.**

David A. Griffin[1], Mike Herzfeld[1], Mark Hemer[1] and Darren Engwirda[2]

[1]Oceans and Atmosphere, CSIRO, Hobart, TAS 7000, Australia
[2]Center for Climate Systems Research, Columbia University, New York City, NY, USA and NASA Goddard Institute for
Space Studies, New York City, NY, USA

*Correspondence to*: David Griffin (David.Griffin@csiro.au)

**Abstract.** While the variations of tidal range are large and fairly well known across Australia (less than 1 m near Perth but more than 14 m in King Sound), the properties of the tidal currents are not. We describe a new regional model of Australian tides and assess it against a validation dataset comprising tidal height and velocity constituents at 615 tide gauge sites and 95 current meter sites. The model is a barotropic implementation of COMPAS, an unstructured-grid primitive-equation model that is forced at the open boundaries by TPXO9v1. The Mean Absolute value of the Error (MAE) of the modelled M2 height amplitude is 8.8 cm, or 12 % of the 73 cm mean observed amplitude. The MAE of phase (10°), however, is significant, so the M2 Mean Magnitude of Vector Error (MMVE, 18.2 cm) is significantly greater. The Root Sum Square over the 8 major constituents is 26% of the observed amplitude.. We conclude that while the model has skill at height in all regions, there is definitely room for improvement (especially at some specific locations). For the M2 major-axis velocity amplitude, the MAE across the 95 current meter sites, where the observed amplitude ranges from 0.1 cm s$^{-1}$ to 156 cm s$^{-1}$, is 6.9 cm s$^{-1}$, or 22 % of the 31.7 cm s$^{-1}$ observed mean. This nationwide average result is encouraging, but it conceals a very large regional variation. Relative errors of the tidal current amplitudes on the narrow shelves of NSW and Western Australia exceed 100 %, but tidal currents are weak and negligible there compared to non-tidal currents, so the tidal errors are of little practical significance. Looking nation-wide, we show that the model has predictive value for much of the 79 % of Australia's shelf seas where tides are a major component of the total velocity variability. In descending order this includes the Bass Strait, Kimberley to Arnhem Land and Southern Great Barrier Reef regions. There is limited observational evidence to confirm that the model is also valuable for currents in other regions across northern Australia. We plan to commence publishing 'unofficial' tidal current predictions for chosen regions in the near future, based on both our COMPAS model and the validation data set we have assembled.

# 1 Introduction

Tidal currents are a major component of the velocity variability for most of the Australian continental shelf, yet tidal current predictions are only listed in the Australian National Tide Tables for 7 sites, 5 of which are in Torres Strait. As part of a project to map Australia's tidal energy resource, and as a step towards an operational, model-based tidal current forecasting ability,





we have compiled a tidal currents harmonic constituents validation dataset at 95 sites based on observations acquired by a number of agencies. This is a significant number of sites, but it is still small compared to the 683 sites for which the Bureau of Meteorology Tidal Unit has estimates of tidal height harmonic constituents. We use these validation datasets for currents and heights to assess the errors of a newly configured barotropic implementation of an unstructured-grid tidal model for the

Australian continental shelf. This tells us how well the tidal component of the total variability can be predicted. Taking non-





tidal currents into account as well, we identify the regions of Australia where model-based tidal current predictions are not only accurate, but also a large part of the total variability.

**Figure 1 Model mesh spacing (km, log scale). Abbreviated names are: CC=Clarence Channel, VanDG=Van Diemen Gulf, GOC=Gulf of Carpentaria, CGBR=Central Great Barrier Reef, SGBR=Southern GBR, SEQ=Southeast Queensland, NSW=New South Wales, Bass=Bass Strait, Tas=Tasmania, Banks=Banks Strait, SA=South Australia, SW=South West. The colour bar tick labels apply also to the bar graph above.**





## 2 Model configuration

We generated time-series of tidal predictions surrounding Australia using the unstructured model COMPAS (Coastal Ocean Model Prediction Across Scales) (Herzfeld et al., 2020). This model was chosen over structured model counterparts due to its capacity for superior resolution placement and transition, allowing high resolution to be placed in areas of interest, and low resolution elsewhere. This significantly reduces the number of cells required to model such a large domain, resulting in an acceptable computational cost. COMPAS is a coastal ocean model designed to be used at scales ranging from estuaries to

regional ocean domains. It is a three-dimensional (3D) finite volume hydrodynamic model based on the 3D equations of momentum, continuity and conservation of heat and salt, employing the hydrostatic and Boussinesq approximations. The equations of motion are discretised on arbitrary polygonal meshes according to the TRiSK numerics (Thuburn et al., 2009; Ringler et al., 2010), which is a generalisation of the standard Arakawa C-grid scheme to unstructured meshes. The horizontal terms in the governing equations (momentum advection, horizontal mixing and Coriolis) are discretised using the TRiSK

numerics, whereas the pressure gradient and vertical mixing are discretised using the finite difference approach outlined by Herzfeld (2006). The horizontal mesh must be an orthogonal, centroidal and well-centred "primal-dual" tessellation, typically consisting of collections of Voronoi cells and their dual Delaunay triangles. The 3D model may operate using "z" or s vertical coordinates; however, in the present application a depth-averaged configuration is used, as a developmental step of a more complete model of Australia's coastal ocean. The bottom topography is represented using partial cells. COMPAS has a

nonlinear free surface and uses mode splitting to separate the two-dimensional (2D) mode from the 3D mode. The model uses explicit time-stepping throughout, except for the vertical diffusion scheme which is implicit.

COMPAS uses the unstructured meshing library JIGSAW (Engwirda, 2017) to generate the underlying unstructured mesh. JIGSAW produces high quality meshes that support the requirements of the TRiSK numerics. The mesh of the model discussed

here was generated using a dual weighting function dependent on bottom depth and a preliminary estimate of the tidal current speed, such that those regions with shallow water and high tidal velocities receive high resolution and vice versa. An initial configuration with resolution depending on tidal height amplitude gave poor results because some straits with strong flows but only moderate height amplitude received only moderate resolution. The mesh has 183,810 2D cells with an indicative cell size ranging from 332 m to 63 km (Fig. 1). Eighty per cent of cells have sizes between 1900 m and 7100 m. The mean length of

edges in the mesh is 3680 m. Note that a regular structured grid covering the same spatial domain at the same mean resolution would require ~1.5 million 2D cells.

The model topography (Fig. 2) uses bathymetry from the Geosciences Australia (2002) database, with regions outside its extent filled using the global database dbdb2 (Naval Research Laboratory Digital Bathymetry Data Base https://www7320.nrlssc.navy.mil/DBDB2_WWW/). This was supplemented with high resolution datasets in the Great Barrier

Reef (Beaman, 2010) and northern Australia (https://ecat.ga.gov.au/geonetwork/srv/eng/catalog.search#/metadata/121620). Onsite depth measurements at the locations (Fig. 2) of the tidal currents validation data discussed below were not used for



estimating the model topography, thus providing a limited but independent validation data set. The minimum depth (at zero tide) in the model is 4 m for most of the grid, but 8 m in the NW, NE and in Gulf St Vincent. Depth was median filtered to remove sharp gradients. A channel of 12 m was manually included in King Sound (in the NW) to correct an obvious error

there. A similar bathymetry correction was also made in Western Port (near Melbourne). These bathymetric changes had significant effect on the local tidal response, and it is anticipated that further model improvement will follow from bathymetry corrections based on observations of the real topography.

**Figure 2 Model depth (m, log scale, spanning just a restricted range). Otherwise like Fig. 1.**





The tide is introduced through eight tidal constituents (M2 S2 N2 K2 K1 O1 P1 Q1) from the TPXO9v1 1/6° global model (Egbert and Erofeeva, 2002; https://www.tpxo.net/) and applied at the open boundary using the condition described by Herzfeld et al. (2020). The Herzfeld et al. (2020) scheme includes a normal and tangential velocity Dirichlet condition with provision for a local flux adjustment on normal velocity to maintain domain-wide volume continuity. Thus, the surface height is not directly constrained at the boundary but is instead computed via volume flux divergence as it is in the model interior.

For the present application, we found that flux adjustments to constrain the sea surface height were not required; prescribing the transports at the boundary was sufficient to achieve the target height. This situation is quite unusual. One necessary step to achieve this was to use the TPXO components of transport on their native (Arakawa C) grid and use the depths in COMPAS to convert the transports to depth-averaged, cell-edge normal velocity, thus compensating for bathymetry differences between our model and the TPXO model. The model was run in 2D mode only, using a time-step of 1 s, achieving a runtime of ~5:1

on twelve processors. A spatially constant bottom drag coefficient of 0.003 was used to compute bottom stress. Tidal potential forcing is optionally applied in the model but we found that it made very little difference (excepting the run time) compared with other parameters such as friction, so we have omitted it for the long (1 year) run of the model described here.

For many test runs of the model, it was started from rest and run for either 7 or 30 days from 24 Feb 2017 including a 1-day ramp period. The model run was assessed against height and velocity observations by comparing it with harmonically

synthesised (using T-Tide v1.3b, Pawlowicz et al., 2002) time series at all sites for which tidal constituents (up to 13) are available (see below). There are very many more such sites than the number of observed time-series available for any particular month, thus providing a more comprehensive assessment.

The model parameters adjusted during the series of test runs included: 1) the bottom drag coefficient, 2) spatial variations of bottom drag,  3) bottom drag scheme, 4) coastal depth, 5) horizontal viscosity, 6) turbulence closure scheme, 7) bathymetry

smoothing, 8) flux adjustment timescale, 9) tidal potential forcing on/off (left off finally), 10) bathymetry data source and  11) interior relaxation to tpxo on/off (left off finally) . These experiments proceeded in an ad-hoc search for closer agreement with the observations. Apart from this 'model tuning', no data assimilation was used with these model runs.

For the model configuration described here, it was run for 365 days from 24 Feb 2017, and then tidally analysed for 13 constituents (M2 S2 N2 K2 K1 O1 P1 Q1 M4 MS4 M6 2MS6 and 2N2) so that 1) its performance can be described for all

those individual constituents, and 2) predictions can be made for any time or place within the domain without having to run the model. The COMPAS model code, the output time series and tidal constituents at all points of the mesh  are freely available, as described in Sections 9 and 10.

**3 Current meter observations**

Acoustic Doppler current profilers (ADCPs) of various types have been deployed more than 1097 times as part of Australia's

Integrated Marine Observing System (IMOS) at 55 sites over the continental shelf around Australia since 2007. The ADCPs are almost all moored within a few metres of the sea bed, and sense the water velocity over the lower 80–85% of the water





column. We have taken the depth-average of these observations, concatenated all records from individual instrument deployments at the same nominal position, and determined the tidal constituents using the UTide software of Codiga (2011). Thirteen constituents (M2 S2 N2 K2 K1 O1 P1 Q1 M4 MS4 M6 2MS6 and 2N2) were analysed at the 64 sites having records

exceeding 180days. The records at other sites were all long enough to resolve 11 constituents (the full list minus K2 and P1). Apart from the deployments off the NW of the continent, these 55 IMOS sites tend to be at locations where tidal currents are not particularly strong. As a means of quantifying the relative magnitude of tidal and sub-tidal depth-average velocity, we determined the principal axis of the subtidal variability (using singular value decomposition) and computed the root mean square (RMS) of the major and minor axis components. Details of the IMOS ADCP deployments are at

http://oceancurrent.imos.org.au/timeseries/ along with regional graphics comparing the tidal and sub-tidal ellipse parameters (as well as the mean velocity for each deployment).

Penesis et al. (2020) give details of ADCP deployments that deliberately sought to observe tidal currents for two of Australia's most prospective tidal energy development regions. These include seven locations in the Clarence Channel near Darwin and seven locations in Banks Strait at the NE tip of Tasmania. We determined tidal velocity constituents, the mean and sub-tidal

ellipse parameters from these data as above.

We have included data from 10 of the sites where Middleton et al. (1984) and Griffin et al. (1987) deployed current meters on the Southern Great Barrier Reef (SGBR, see Fig. 2) in order to study both the anomalous tides and the sub-tidal variability. These observations were made by single, mechanical RCM4 Aanderaa current meters with several drawbacks compared to ADCPs. Due to limited storage capacity, the flow direction was only sampled instantaneously once an hour, so short-period

changes of direction were not averaged. To minimise noise due to waves, the instruments were moored fairly low in the water column (typically 7 m off the seabed), thereby probably underestimating the depth-average velocity. Some had to be deployed close to islands, with the result that they recorded effects (such as asymmetric ebb and flood directions) that the model is unlikely to be able to reproduce at specific locations due to its imperfect representation of topography. Nevertheless, we have included these  records in our validation dataset, processed as above, despite the quality questions because 1) the tides in this

region are important for navigation (e.g. through Hydrographers Passage), and 2) in the hope that future models with finer meshes and better topography may be able to better distinguish observation error from model error.

Lastly, we also extracted 13 current meter records from the CSIRO archives (https://www.cmar.csiro.au/data/trawler/), choosing sites in Bass Strait, the NW shelf and the Gulf of Carpentaria where tidal currents are significant. These were mostly point measurements, either by acoustic or mechanical (Aanderaa) current meters. Where two instruments were deployed on a

mooring, we simply averaged the data for the period when both were operating.

In support of this paper and future studies of the tides of Australia, we have published this validation data set as a netCDF file containing up to 13 tidal constituents, and the subtidal statistics, for each of the 95 locations discussed above (see Section 10).





## 4 Tide gauges

The National Operations Centre (NOC) Tidal Unit of the Bureau of Meteorology
(http://www.bom.gov.au/oceanography/projects/ntc/ntc.shtml) kindly provided 8 tidal height constituents (M2 S2 N2 K2 K1 O1 P1 Q1) for 683 sites, of which 626 are within the COMPAS domain. To this we have added nine sites from the UNSW SGBR dataset bringing the total to 635 before applying quality control.

## 5 Model-data comparison method

The model-data comparisons presented in this paper are based on the tidal constituents (M2 S2 N2 K2 K1 O1 P1 Q1)
determined from the model and observational time-series (rather than the time series approach used during model tuning) for all the usual reasons. We focus on results for M2, or sums over the 8 major constituents. Availability of the full set of model-data comparisons for 13 constituents, 18 regions and 5 variables is covered in Section 10.

### 5.1 Tide gauges

When comparing the model with tide gauges, we select the closest model grid point if one exists within 11 km. We calculate
the model error (model minus observation) for amplitude and phase individually as well as the vector error (taking both phase and amplitude into account) for each tidal constituent. Summing over a number of sites within a certain geographic region, we then compute the Mean of the Absolute value of the amplitude Error (MAE), the Mean Magnitude of Vector Error (MMVE), the mean of the amplitude error and the mean of the observed amplitude (for expressing the MAE or MMVE as a relative error or RE). We use MAE and MMVE in preference to root-mean-squared errors because the MAE and MMVE are
less affected by outliers. Outliers are a significant issue, as we will discuss below with reference to Table 1, which lists the sites we have chosen to exclude from the tidal heights dataset. We combine analyses across constituents by computing the Root Sum of Squared (RSS) MAEs and MMVEs. In order to estimate the total regional-mean tidal relative error, we also compute the RSS of the area-mean observed amplitudes. These statistics are computed for a number of regions (bounding boxes are shown in Fig. 1) around Australia as well as for the entire country and listed in Table 2. We have not attempted to
account for the uneven distribution of the data points around Australia, other than to compute regional means as well as the nationwide means. Nor have we attempted to estimate errors of the observational tidal constituents based on factors such as record length or instrument type, these being unknown in many cases.

### 5.2 Current meters

When comparing with current meters, we select the grid point for which a penalty function $J=D/(5C)+|H_m-H_o|/H_o$ is
minimised, where D are the distances to the model grid point, C are the sizes of the cells, $H_m$ are the model depths and $H_o$ is the onsite depth at the observation point. This is an attempt to mitigate the effect of the model's imperfect topography, by finding the nearest depth-matching (if possible) model counterpart of the observation. We then proceed as for tide gauges, but





with the amplitude and phase of the major axis velocity taking the place of height. Errors of the major axis inclination and minor axis amplitude are shown graphically and are listed in Table 3 but are not otherwise included. Three sorts of site-specific

relative error are listed in Table 3: 1) the M2 major axis velocity amplitude error relative to the observed amplitude $reM2 = (|maj_m| - |maj_o|)/|maj_o|$ , 2) the M2 major axis velocity vector error relative to the observed amplitude $re\overline{M2} = |maj_m - maj_o|/|maj_o|$ , and 3) reLF, which has the observed sub-tidal ('low frequency') RMS major axis velocity $sub\_o$ included in both numerator and denominator. The first two measures characterise the model's ability to do what it is designed for, which is just to simulate tides. The first of these is for users who need to know tidal range but not at any particular time.

The second is for applications where timing is also important. The third acknowledges that tides are not the dominant component of velocity variability everywhere. Using a tidal model alone (i.e. without a model of other processes) to predict the total current (characterised by $maj\_o+sub\_o$) will result in an error determined by $sub\_o$ if the tidal error is zero. Where tidal and sub-tidal variability are equal, the upper limit of reLF is 50%.

Table 3 lists sites by ascending reLF, and includes averages of the sites with lowest, middle and greatest reLF, for most

columns. For the 'm-o' column the average is mathematically an MAE, but with a non-geographic sample of sites. Table 4 is like Table 2, with major axis velocity amplitude and phase taking the place of height amplitude and phase, for the same 8 constituents.

# 6 Results

## 6.1 Tidal height

Since we have no reliable, objective (model independent) way of knowing which tide gauge observations (or more precisely, the analysed tidal constituents) are more accurate than others, we have cautiously employed a largely model-based quality control procedure. This procedure excludes sites if:

- The absolute value of M2 error exceeds 20 cm and an observed M2 amplitude within 10 km is less by more than 20 cm (excludes four sites)

- The observed amplitude is less than 4 cm (two sites)

- The observed amplitude exceeds 10 cm and is less than half, or more than twice the model amplitude (14 sites)

- The observed and modelled phase differ by more than 90° (six sites).

**Table 1: Blacklisted tide gauges. Tests are on the nearest neighbour difference (cm), the observed M2 amplitude (cm) and the model**
**M2 amplitude (cm) and phase relative to the observed values.**

| Site# | Site | Latitude | Longitude | nndiff | Observed | Model | Phase diff |
|---|---|---|---|---|---|---|---|
| 67 | Kai-Maituine Reef - Northeast | 10.23S | 143.15E | 0 | 69 | 61 | 94 |
| 71 | Dauan Island | 9.411S | 142.54E | -7 | 31 | 14 | 17 |





| | | | | >20cm | <4cm | o*0.5, o*2 | >90 ° |
|---|---|---|---|---|---|---|---|
| 105 | Sharp Point | 10.97S | 142.72E | -49 | 23 | 92 | -39 |
| 125 | Harvey Island | 11.97S | 143.27E | -44 | 19 | 75 | 0 |
| 152 | Endeavour River North | 15.43S | 145.2E | -22 | 31 | 59 | -11 |
| 187 | Rib Reef | 18.47S | 146.87E | 0 | 22 | 69 | -9 |
| 333 | South Channel | 38.3S | 144.71E | -5 | 21 | 10 | 27 |
| 378 | Maatsuyker Island | 43.67S | 146.32E | 0 | 23 | 8 | 14 |
| 457 | Nornalup Inlet | 35S | 116.73E | 0 | 2 | 6 | -51 |
| 465 | Mandurah | 32.53S | 115.72E | 0 | 3 | 5 | -15 |
| 490 | Monkey Mia | 25.8S | 113.72E | 0 | 38 | 10 | 12 |
| 577 | Bonaparte Gulf | 12.83S | 128.47E | 0 | 14 | 82 | -137 |
| 586 | Catfish Island | 14S | 129.48E | 86 | 268 | 172 | -46 |
| 631 | Peacock Island | 11.02S | 132.45E | 0 | 19 | 68 | 18 |
| 659 | Mallison Island | 12.18S | 136.1E | 0 | 173 | 14 | 88 |
| 668 | Centre Island | 15.75S | 136.81E | 0 | 40 | 18 | 37 |
| 669 | Mornington Island | 16.67S | 139.17E | 0 | 14 | 7 | 18 |
| 672 | Albert River Mouth | 17.55S | 139.76E | 0 | 20 | 13 | 121 |
| 674 | Sweers Island | 17.11S | 139.59E | 0 | 15 | 6 | 112 |
| 675 | Karumba | 17.49S | 140.83E | 0 | 17 | 18 | 90 |
| | Failure criterion | | | **>20cm** | **<4cm** | **o*0.5, o*2** | **>90 °** |
| | Number of failures | | | **4** | **2** | **13** | **5** |

With the 20 sites listed in Table 1 excluded, the M2 MAE across 615 sites is 8.8 cm (Table 2), or 12 % of the mean observed amplitude, which is 72.5 cm. The resulting scatter plot (Fig. 3, note the log-log axes) of model vs observed height amplitude

still has points that could be considered outliers; at 5 % of sites the negative errors are ~3 to 10 times the MAE. But we have not excluded these along with the other 20, for lack of clear evidence that they are due to observation error rather than model error.

The nation-wide bias is small (-0.6 cm, see Table 2), but some regional biases are not. The region with the biggest M2 bias (-8.8 cm) is clearly (see Table 2) the Southern Great Barrier Reef, where the model underpredicts the large tides within about

100 km of the head of Broad Sound

The region with the biggest M2 amplitude MAE (at 17.9cm) is the one we abbreviate here as 'Arnhem' (rather than Joseph Bonaparte Gulf and Arnhem Land) but across this region there is a mix of under and over-prediction. The modelled M2 height amplitude is too small in Van Diemen Gulf and the head of Joseph Bonaparte Gulf but too great at many of the offshore sites where the observed amplitude is small.

There are large M2 phase errors (Fig. 4) at many sites. While some are possibly due to observation error, the predominance of positive phase errors at locations of strong tides points to a problem in the model. The region with the biggest M2 phase MAE is the Kimberley (18°) (Table 2), nearly twice the all-site average of 10.4°. The significant phase errors are why the Australia-wide M2 MMVE (18.2 cm) is so much greater than the M2 MAE (8.8 cm).



The next most energetic constituent after M2 (72.5 cm averaged across all sites) is S2 (35.7 cm). S2 has the next-greatest

MMVE (11.4 cm, because of large phase errors in the Kimberley).

Summing over 8 constituents, and taking both phase and amplitude errors into account, the RSS MMVE across all sites is 23.9 cm, or 26.4 % of the mean observed amplitude. The three regions with the lowest relative error (13, 15 and 16 %) are Central Great Barrier Reef, New South Wales and the South West, while the regions with the highest (31-36%) are South Australia, the wide shallow seas in the tropics: Torres Strait, Joseph Bonaparte Gulf and Arnhem Land, the Kimberley and Gulf of

Carpentaria. Thus, the greatest regional-average relative errors of modelled height are about twice the size of the least. Both are small enough to conclude that the model has skill, but large enough to conclude that there is still room for improvement.







**Figure 3 M2 height amplitude as a colour-fill map (the model) and points (observations), and inset as a quantity-quantity plot. Statistics listed are percentiles of 1) the whole model height field, 2) m=model at validation sites, 3) model error m-o and 4) o=observed values. <|m-o|> is the Mean of the Absolute value of m-o. <m-o> is the mean error, or bias. <m> and <o> are the mean modelled and observed amplitudes. A log scale is used, starting at 10cm, so not all points can be shown.**





**Figure 4 M2 height phase (otherwise like Fig. 3, except the y-axis of the inset is the phase error rather than phase).**





**Table 2: Tidal height and phase region-average statistics, for eight constituents (and their root sum of squares).**
**Height (cm)**
**mean observed amplitude < o >**

|        | Aust  | Arnhem | GOC  | Torres | CGBR | SGBR  | SEQ  | NSW  | Bass | Tas  | SA   | SW   | Pilbara | Kimb. |
|--------|-------|--------|------|--------|------|-------|------|------|------|------|------|------|---------|-------|
| #sites | 615   | 78     | 111  | 66     | 59   | 67    | 29   | 27   | 54   | 24   | 62   | 31   | 41      | 43    |
| M2     | 72.5  | 112.1  | 59   | 60.4   | 56.5 | 112.4 | 59.7 | 46.5 | 56.7 | 46.5 | 25.5 | 6.6  | 77.5    | 168.3 |
| S2     | 35.7  | 50.1   | 34.3 | 40.8   | 33   | 42.2  | 17.5 | 11.1 | 12.1 | 7    | 26.7 | 7    | 44.4    | 99.5  |
| N2     | 16.2  | 21.7   | 18.1 | 20.9   | 18.6 | 27.7  | 12.3 | 10.5 | 12.2 | 11.2 | 1.9  | 2.1  | 12.7    | 27.1  |
| K2     | 10    | 14.1   | 9.5  | 11     | 9.2  | 12.2  | 5.1  | 3.3  | 2.8  | 2    | 7.8  | 2.1  | 11.6    | 28.2  |
| K1     | 29.6  | 42.2   | 42   | 47     | 31.4 | 31.9  | 18.9 | 15   | 15.9 | 17.3 | 24.2 | 17.5 | 21.2    | 31.6  |
| O1     | 17.7  | 27.1   | 24.5 | 23.9   | 15.1 | 16.4  | 10.6 | 9.4  | 10.9 | 12   | 16.5 | 12.6 | 13.6    | 19.3  |
| P1     | 8.7   | 11.7   | 12.2 | 13.7   | 9.4  | 9.6   | 5.3  | 4.5  | 5    | 5.6  | 7    | 5.4  | 6.3     | 9.1   |
| Q1     | 3.8   | 6.3    | 4.5  | 4.3    | 2.8  | 3.1   | 2.2  | 2.2  | 2.7  | 3    | 3.7  | 3.1  | 3.2     | 4.6   |
| RSS    | 90.4  | 135.8  | 87.2 | 94.1   | 77.6 | 129.2 | 67.5 | 52.4 | 62.7 | 53.2 | 48.4 | 24.6 | 94.6    | 203.1 |


**mean magnitude of vector error (MMVE)**

|        | Aust  | Arnhem | GOC  | Torres | CGBR | SGBR | SEQ  | NSW  | Bass | Tas  | SA   | SW   | Pilbara | Kimb. |
|--------|-------|--------|------|--------|------|------|------|------|------|------|------|------|---------|-------|
| #sites | 615   | 78     | 111  | 66     | 59   | 67   | 29   | 27   | 54   | 24   | 62   | 31   | 41      | 43    |
| M2     | 18.2  | 32     | 16.8 | 17.3   | 8.5  | 20.4 | 17.6 | 6.5  | 12.6 | 7.7  | 9.1  | 1.6  | 21.9    | 50.1  |
| S2     | 11.4  | 18.7   | 13   | 18.2   | 3.5  | 8.5  | 6.6  | 2.2  | 3.5  | 2.9  | 10.5 | 1.2  | 15.5    | 37    |
| N2     | 4.5   | 6.9    | 5.7  | 7      | 3.2  | 6.5  | 3.7  | 1.9  | 3.1  | 2.3  | 0.94 | 0.53 | 3.4     | 9.7   |
| K2     | 3.4   | 5.2    | 4.1  | 5.6    | 0.86 | 3.4  | 1.8  | 0.58 | 0.95 | 0.88 | 3    | 0.44 | 3.6     | 10.4  |
| K1     | 7.1   | 15.7   | 13.9 | 17.5   | 2.5  | 2.9  | 4    | 2.8  | 3    | 3.6  | 5.3  | 2.5  | 4.9     | 6.2   |
| O1     | 4.2   | 9.3    | 8.5  | 9.8    | 1    | 1.6  | 2.2  | 1.6  | 1.9  | 2.1  | 3.3  | 1.9  | 3       | 3.9   |
| P1     | 2.3   | 4.5    | 4.6  | 5.9    | 0.9  | 0.99 | 1.6  | 0.95 | 1    | 1.3  | 1.6  | 1.1  | 1.5     | 1.8   |
| Q1     | 1.3   | 2.2    | 2.2  | 2.5    | 0.72 | 0.76 | 0.58 | 0.39 | 0.63 | 0.73 | 0.94 | 0.73 | 0.91    | 1.7   |
| RSS    | 23.9  | 42.5   | 28.1 | 34     | 10.2 | 23.6 | 19.9 | 7.9  | 14   | 9.7  | 15.7 | 4    | 28      | 64.3  |
| %obs   | 26.4  | 31.3   | 32.2 | 36.1   | 13.1 | 18.2 | 29.4 | 15.2 | 22.3 | 18.2 | 32.4 | 16.3 | 29.6    | 31.7  |

**mean absolute value of error <|m-o|> (MAE)**

|        | Aust  | Arnhem | GOC  | Torres | CGBR | SGBR | SEQ  | NSW  | Bass | Tas  | SA   | SW   | Pilbara | Kimb. |
|--------|-------|--------|------|--------|------|------|------|------|------|------|------|------|---------|-------|
| #sites | 615   | 78     | 111  | 66     | 59   | 67   | 29   | 27   | 54   | 24   | 62   | 31   | 41      | 43    |
| M2     | 8.8   | 17.9   | 9.1  | 8.3    | 6.3  | 11   | 6.8  | 4.7  | 7.8  | 3.3  | 7    | 0.7  | 5.5     | 10.8  |
| S2     | 5.4   | 9.5    | 7.8  | 10.7   | 2.3  | 5.3  | 3.1  | 1.7  | 2.4  | 1.4  | 7.5  | 0.64 | 3.2     | 7.6   |
| N2     | 2.5   | 4.4    | 3.1  | 3.6    | 2    | 4.2  | 1.8  | 1.2  | 2.1  | 0.94 | 0.53 | 0.38 | 1.2     | 3.4   |
| K2     | 1.7   | 2.7    | 2.1  | 3      | 0.48 | 2.5  | 0.77 | 0.48 | 0.69 | 0.41 | 2.1  | 0.24 | 0.92    | 2.2   |
| K1     | 3.5   | 4.6    | 8.3  | 9.5    | 1.6  | 1.5  | 2.1  | 2.2  | 2.4  | 2    | 2.8  | 1.3  | 1.9     | 2.8   |
| O1     | 2     | 3.2    | 3.8  | 3.3    | 0.73 | 1.1  | 0.93 | 1.1  | 1.4  | 1.2  | 1.6  | 0.88 | 1.3     | 1.9   |
| P1     | 1.2   | 1.5    | 2.5  | 2.9    | 0.61 | 0.52 | 0.95 | 0.77 | 0.72 | 0.89 | 0.87 | 0.57 | 0.61    | 0.91  |
| Q1     | 0.67  | 0.99   | 1.1  | 1.1    | 0.46 | 0.44 | 0.28 | 0.22 | 0.39 | 0.24 | 0.52 | 0.53 | 0.45    | 1.1   |
| RSS    | 11.6  | 21.7   | 15.8 | 17.8   | 7.2  | 13.3 | 8.1  | 5.8  | 8.9  | 4.5  | 11   | 2    | 7       | 14.3  |
| %obs   | 12.8  | 16     | 18.2 | 18.9   | 9.3  | 10.3 | 12.1 | 11.1 | 14.2 | 8.4  | 22.7 | 8.2  | 7.4     | 7     |





**mean error < m-o > (bias)**

|       | Aust | Arnhem | GOC  | Torres | CGBR  | SGBR  | SEQ  | NSW  | Bass | Tas   | SA    | SW    | Pilbara | Kimb. |
|-------|------|--------|------|--------|-------|-------|------|------|------|-------|-------|-------|---------|-------|
| #sites| 615  | 78     | 111  | 66     | 59    | 67    | 29   | 27   | 54   | 24    | 62    | 31    | 41      | 43    |
| M2    | -0.6 | -2     | 1.2  | 1.9    | 5.8   | -8.8  | 5.7  | 4.1  | -5.1 | -0.59 | -2.4  | -0.48 | 1.6     | 0.34  |
| S2    | -1.4 | 0      | -5.9 | -9.8   | 1.7   | -4.3  | 2.7  | 1.6  | -1.5 | -0.63 | 0.31  | -0.49 | 0       | -1.2  |
| N2    | -0.96| -1.8   | -1.6 | -2.7   | 0.89  | -3.2  | 1.5  | 0.91 | -1.5 | -0.25 | 0.34  | -0.2  | -0.11   | -1.8  |
| K2    | -0.41| 0.42   | -0.98| -2.1   | -0.22 | -2.4  | 0.64 | 0.48 | 0    | -0.11 | 0.21  | 0     | 0       | -0.79 |
| K1    | -0.66| 2.4    | -7.4 | -8.9   | 0.36  | 0.31  | 1.7  | 2    | 0.95 | 1     | -1.2  | 0.34  | 0.64    | 1.4   |
| O1    | -0.21| 1.6    | -2.9 | -2.2   | 0.2   | -0.7  | 0.6  | 0.85 | 0.59 | 0.45  | -0.63 | 0.28  | 0.44    | 0.8   |
| P1    | -0.2 | 0.6    | -2.1 | -2.6   | 0     | 0     | 0.94 | 0.69 | 0.22 | 0     | -0.27 | 0.29  | 0       | 0.57  |
| Q1    | -0.2 | 0      | -0.83| -0.9   | -0.16 | -0.19 | 0    | 0.14 | 0    | 0     | 0     | 0     | -0.12   | -0.14 |

**Height phase (°)**
**mean absolute value of error <|m-o|> (MAE)**

|       | Aust | Arnhem | GOC  | Torres | CGBR | SGBR | SEQ  | NSW | Bass | Tas  | SA   | SW  | Pilbara | Kimb. |
|-------|------|--------|------|--------|------|------|------|-----|------|------|------|-----|---------|-------|
| #sites| 615  | 78     | 111  | 66     | 59   | 67   | 29   | 27  | 54   | 24   | 62   | 31  | 41      | 43    |
| M2    | 10.4 | 11.2   | 12.8 | 12.9   | 4.8  | 6.8  | 13.5 | 4.6 | 9.2  | 8.5  | 11.6 | 9.5 | 15.1    | 17.9  |
| S2    | 13.2 | 15.8   | 18.7 | 25     | 3.7  | 7.2  | 15.6 | 6   | 11.7 | 20.2 | 13.8 | 5.9 | 17.1    | 22.5  |
| N2    | 12.6 | 13.3   | 15.8 | 16.2   | 6.4  | 8.1  | 12.8 | 6.2 | 11.1 | 12.7 | 17.3 | 10  | 14.4    | 21.2  |
| K2    | 14.4 | 15.5   | 25.4 | 31.8   | 3.7  | 8.9  | 14.4 | 4.5 | 11.2 | 20.7 | 13.7 | 7.7 | 16.8    | 22.5  |
| K1    | 9.2  | 16.9   | 13.8 | 17.8   | 2.9  | 3.4  | 9    | 5.4 | 5.6  | 9    | 9.2  | 6.5 | 10.3    | 9.1   |
| O1    | 9.2  | 16.4   | 16   | 21.7   | 2    | 3.2  | 9.5  | 6.6 | 5    | 7.1  | 8.8  | 7.1 | 9.5     | 9.3   |
| P1    | 10.3 | 17     | 16.6 | 20.5   | 3.5  | 4.1  | 10.4 | 5.6 | 7.4  | 9.1  | 9.7  | 8.9 | 11.8    | 8.6   |
| Q1    | 13.8 | 15.9   | 25.1 | 31.4   | 10   | 9.9  | 10.8 | 7.5 | 9.7  | 11.4 | 10.5 | 8.4 | 12.6    | 14    |

**mean error < m-o > (bias)**

|       | Aust | Arnhem | GOC  | Torres | CGBR  | SGBR | SEQ  | NSW  | Bass  | Tas   | SA   | SW    | Pilbara | Kimb. |
|-------|------|--------|------|--------|-------|------|------|------|-------|-------|------|-------|---------|-------|
| #sites| 615  | 78     | 111  | 66     | 59    | 67   | 29   | 27   | 54    | 24    | 62   | 31    | 41      | 43    |
| M2    | 2.7  | 7.5    | 5.6  | 7.8    | -4.2  | 3.7  | -8.8 | -4.2 | 5.1   | 3.9   | -3.1 | -0.64 | 7.2     | 13    |
| S2    | 2.7  | 7.2    | 3.7  | 5      | -2    | 2.5  | -9.2 | -5.4 | 5.5   | 2.5   | -1.6 | -4.1  | 11.2    | 15.9  |
| N2    | 3    | 7.4    | 5.6  | 9.2    | -5    | 3.2  | -9   | -3.2 | 5.7   | 5.5   | 2.1  | -1.1  | 4.3     | 14.9  |
| K2    | 1.2  | 5.2    | -7.4 | -9.2   | 0.55  | 5    | -5   | -3.4 | 5.3   | 8.7   | -1.5 | -0.22 | 6.6     | 15.7  |
| K1    | 4.4  | 16.1   | 8.5  | 16.4   | -2.5  | 2.5  | -6.2 | -3.6 | 0.67  | 2.4   | 2.2  | 2.5   | 7       | 6.3   |
| O1    | 5.4  | 15.8   | 14.1 | 21.4   | -0.97 | 2    | -6.9 | -4.6 | 0     | 3.2   | 2.7  | 3     | 6.7     | 6.9   |
| P1    | 4.5  | 15     | 10   | 19.3   | -2.9  | 2.5  | -5.9 | -4.2 | -0.34 | 2     | 3.4  | 0.78  | 8.3     | 4.7   |
| Q1    | 6.4  | 11.7   | 21.8 | 31     | -0.12 | 7.2  | -7.7 | -4.7 | 3.5   | -0.33 | 2    | -0.3  | 5.5     | 0.82  |



**Figure 5 Amplitude of the M2 major axis velocity, otherwise like Fig. 3. Black (model, at a random subset of grid points) and magenta (observed) velocity ellipses use the scale shown.**

## 6.2 Tidal currents

Perhaps the most striking difference between maps of the M2 major axis amplitude (Fig. 5) and the M2 height amplitude (Fig. 3) is that the currents have more small-scale variability, clearly associated with the local topography, as well as the regional variability that broadly reflects the regional variations of tidal range. Characterising and analysing the distribution of the errors





as well as the signal is not straightforward, but is what we will attempt to do, after looking at some of the site-specific results listed in Table 3.

The first line of Table 3 is for the IMOS site north of Heron Island in the Southern Great Barrier Reef. It is the first line because it has the lowest reLF, which in turn is because the errors of the M2 major axis velocity phase and amplitude are both small (-1° and -3 cm s$^{-1}$), while the amplitude of the observed M2 tidal currents is large (50 cm s$^{-1}$) compared to the rms sub-tidal velocity (8 cm s$^{-1}$). Site CW3 (line 3) sampled by Penesis et al. (2020) in Banks Strait is a more energetic site but the errors of the major axis velocity phase and amplitude are both relatively small (9° and 1 cm s$^{-1}$) nevertheless. It is also a tidally dominated site, (98 cm s$^{-1}$ for M2 compared to the sub-tidal velocity of just 7 cm s$^{-1}$). As it happens, the error of the minor axis is also very small (both are essentially zero) here, but the error of the inclination is not (-28°T observed but -52°T modelled). Site CW1 (line 6) is about 3 km away (just one grid cell) and has a greater amplitude error (14 cm s$^{-1}$) but less inclination error (2°). Looking down the table we see that 8 of the 18 lowest-error sites are in Banks Strait. This is clearly a region where the model in its present form is capable of producing current velocity predictions with low relative error, so is the first to be discussed in the next section.

At the other extreme (at the bottom of Table 3) is GBRLSL, a site off the Great Barrier Reef in 330 m of water where the observed M2 major axis velocity is essentially zero, but the model estimate is 7 cm s$^{-1}$. Second-bottom is NRSNIN, an IMOS ADCP at the Ningaloo Reef National Reference Site in Western Australia, where the observed M2 major axis amplitude is just 7 cm s$^{-1}$ while the model estimate is 20 cm s$^{-1}$. From the prediction point of view, the errors at these 2 sites are compounded by there being fairly strong (12 and 18 cm s$^{-1}$) sub-tidal currents, but small mean current (4cm/s). One thing these two sites have in common is that they are over steep topography where sharp gradients are common, so part of the poor agreement is bound to be due to representation error (that error that occurs when you compare a point measurement with an area-average). But even so, these are probably not sites where tidal predictions will be of much practical use.

Table 3 includes statistics that characterise model error averaged over sites grouped according to whether reLF is in the lowest, middle and highest third. The MAE over this first third is 7 cm s$^{-1}$ (an 11 % average relative error), while the MMVE is 14 cm s$^{-1}$, a 21 % average relative error or 29 % if sub-tidal currents are taken into account as well. For the locations that these sites are representative of, you could argue that the tidal model is not only useful, but is enough by itself, i.e. a short-term forecast of sub-tidal current velocity would not often make a significant contribution (since its mean rms value is around 6 cm s$^{-1}$, just 10% of the mean M2 amplitude). For the middle group the average M2 tidal current amplitude (27 cm s$^{-1}$) alone still exceeds the sub-tidal variability (10 cm s$^{-1}$), but the dominance is less than for the first third and the errors (MMVE=12 cm s$^{-1}$) of the tidal model are not insignificant. The average reLF for this group is 59 %, which could be argued as being acceptable, but with there being much room for reduction, either by improvements to the tidal model or addition in near-real time of a skilful forecast of sub-tidal variability. For the final third, the observed tidal currents are mostly insignificant (3 cm s$^{-1}$ compared to 22 cm s$^{-1}$), so it doesn't really matter what the predicted tidal velocity is, as long as it is weak. This last group includes all 11 sites in New South Wales and south-east Queensland regions, five of the deeper (~100 m or more) sites in South Australia,





and all eight of the sites in south-west Western Australia. We will now look more closely at the regions where tidal currents

are a large fraction of the variability.

**Table 3: Model errors at current meter sites - M2 constituent**

**Columns list: current meter site name and location then 3 measures of the observed, depth-averaged, non-tidal velocity: |mean|, dir and sub_o, which are the magnitude and (compass) direction of the mean, and the magnitude of the root mean square of the sub-**
**tidal low-pass filtered velocity. Next, observed (_o) and model (_m) values of depth h, M2 major axis inclination inc, minor and major axis amplitudes min and maj. Next, errors maj_m-maj_o and g_m-g_o (m-o for short) of the major axis amplitude and Greenwich phase g, then the magnitude of the vector (amplitude and phase) error |m̃-õ|. Next, 3 types of M2 percentage relative errors: reM2 = (m-o)/o, reM̃2 = |m̃-õ|/o; and reLF =(|m̃-õ|+sub_o)/(o+sub_o). Sites are listed by ascending reLF. The means (over successive thirds of the dataset, and then for all of it) of the absolute value of some quantities are given. Note that observed inclination**
**angles are chosen to be -90°T to 90°T. Listed model inclinations and Greenwich phases are both flipped 180° in a few sensible instances.**

| Row | Site | Region | lat. | long. | \|mean\| | dir | sub_o | h_o | h_m | inc_o | inc_m | min_o | min_m | maj_o | maj_m | m-o | m-o | \|m̃-õ\| | reM2 | reM̃2 | reLF |
|---|---|---|---|---|---|---|---|---|---|---|---|---|---|---|---|---|---|---|---|---|---|
| | | | °S | °E | cm s⁻¹ | °T | cm s⁻¹ | m | m | °T | °T | cm s⁻¹ | cm s⁻¹ | cm s⁻¹ | cm s⁻¹ | Δ° | cm s⁻¹ | % | % | % |
| 1 | GBRHIN | SGBR | 23.38 | 151.99 | 3 | -33 | 8 | 45 | 41 | 78 | 81 | 8 | 8 | 50 | 47 | -3 | -1 | 3 | -6 | 6 | 19 |
| 2 | ITFFTB | Arnhem | 12.29 | 128.48 | 4 | 121 | 6 | 108 | 105 | -52 | -55 | -7 | -7 | 35 | 38 | 2 | 0 | 2 | 7 | 7 | 20 |
| 3 | CW3 | Bass | 40.55 | 148.08 | 4 | 115 | 7 | 33 | 31 | -28 | -52 | 0 | 0 | 98 | 99 | 1 | 9 | 15 | 1 | 15 | 21 |
| 4 | NRSDAR | Arnhem | 12.34 | 130.71 | 3 | 79 | 4 | 18 | 16 | -60 | -60 | 7 | 8 | 55 | 62 | 7 | 6 | 9 | 13 | 17 | 22 |
| 5 | Darwin_C3 | Arnhem | 12.07 | 131.02 | 7 | 88 | 5 | 56 | 30 | 89 | 79 | -2 | 0 | 118 | 100 | -19 | 7 | 23 | -16 | 19 | 22 |
| 6 | CW1 | Bass | 40.53 | 148.06 | 0 | 90 | 0 | 32 | 30 | -52 | -54 | -2 | 1 | 82 | 96 | 14 | 8 | 19 | 17 | 23 | 23 |
| 7 | CW4A1 | Bass | 40.67 | 148.09 | 6 | 84 | 9 | 30 | 32 | -71 | -70 | -3 | -1 | 133 | 128 | -5 | 10 | 23 | -4 | 18 | 23 |
| 8 | CW2A1 | Bass | 40.58 | 148.1 | 9 | 121 | 11 | 44 | 33 | -50 | -53 | -3 | -1 | 123 | 123 | 0 | 10 | 21 | 0 | 17 | 24 |
| 9 | DARBGF | Arnhem | 12.11 | 130.59 | 1 | -47 | 1 | 30 | 30 | -89 | -89 | 6 | 4 | 56 | 65 | 9 | 9 | 12 | 15 | 22 | 24 |
| 10 | CWTb1 | Bass | 40.68 | 148.23 | 16 | 128 | 9 | 63 | 45 | -68 | -69 | -3 | -2 | 87 | 99 | 12 | 6 | 15 | 14 | 17 | 25 |
| 11 | BASS-CS91 | Bass | 40.14 | 144.25 | 3 | 42 | 10 | 53 | 50 | 46 | 47 | 7 | 7 | 58 | 52 | -6 | 6 | 8 | -11 | 14 | 27 |
| 12 | North Rf | SGBR | 23.16 | 151.96 | 4 | 2 | 7 | 62 | 58 | -75 | -81 | 4 | 3 | 44 | 46 | 1 | 9 | 7 | 3 | 16 | 28 |
| 13 | CW4A2 | Bass | 40.73 | 148.34 | 7 | 87 | 5 | 36 | 36 | -72 | -74 | 14 | 11 | 66 | 69 | 3 | 13 | 15 | 4 | 23 | 28 |
| 14 | Darwin_CTbW | Arnhem | 12.02 | 130.97 | 9 | 234 | 2 | 22 | 22 | 65 | 86 | -1 | -2 | 89 | 96 | 6 | 15 | 25 | 7 | 28 | 29 |
| 15 | BASS-CS91 | Bass | 39.5 | 148.01 | 6 | 137 | 4 | 47 | 42 | 61 | 75 | 11 | 14 | 50 | 61 | 11 | 5 | 12 | 22 | 24 | 29 |
| 16 | C1A3 | Bass | 40.69 | 148.12 | 12 | 12 | 8 | 27 | 25 | -75 | -67 | -1 | -1 | 144 | 120 | -24 | 12 | 37 | -17 | 26 | 30 |
| 17 | KIM200 | Kimberley | 15.53 | 121.24 | 5 | 241 | 9 | 208 | 215 | -59 | -56 | 7 | 9 | 22 | 21 | 0 | 0 | 0 | -1 | 1 | 30 |
| 18 | CW2A2 | Bass | 40.7 | 148.2 | 12 | 156 | 7 | 44 | 39 | -38 | -60 | -2 | -2 | 85 | 95 | 10 | 11 | 20 | 12 | 24 | 30 |
| 19 | KIM100 | Kimberley | 15.68 | 121.3 | 5 | 213 | 11 | 99 | 96 | -49 | -53 | 13 | 14 | 40 | 41 | 1 | 6 | 4 | 3 | 11 | 30 |
| 20 | GBRHIS | SGBR | 23.51 | 151.96 | 2 | 40 | 4 | 47 | 45 | 89 | 84 | 2 | 5 | 32 | 39 | 7 | 3 | 7 | 21 | 22 | 31 |
| 21 | KIM050 | Kimberley | 16.39 | 121.59 | 3 | 257 | 8 | 59 | 56 | -72 | -73 | 26 | 29 | 44 | 49 | 6 | 8 | 9 | 13 | 20 | 32 |


| Row | Site | Region | lat. | long. | \|mean\| | dir | sub_o | h_o | h_m | inc_o | inc_m | min_o | min_m | maj_o | maj_m | m-o | m̄-o | \|m̄-ō\| | reM2 | reM̄2 | reLF |
|---|---|---|---|---|---|---|---|---|---|---|---|---|---|---|---|---|---|---|---|---|---|
| | | | °S | °E | cm s⁻¹ | °T | cm s⁻¹ | m | m | °T | °T | cm s⁻¹ | cm s⁻¹ | cm s⁻¹ | cm s⁻¹ | cm s⁻¹ | Δ° | cm s⁻¹ | % | % | % |
| 22 | C1A1 | Bass | 40.67 | 148.24 | 14 | 130 | 8 | 56 | 42 | -75 | -76 | -2 | -1 | 84 | 97 | 14 | 10 | 21 | 16 | 25 | 32 |
| 23 | ARA-GOC87 | GOC | 10.64 | 136.94 | 7 | -42 | 3 | 57 | 58 | -86 | -80 | 6 | 2 | 21 | 25 | 4 | -7 | 5 | 19 | 23 | 32 |
| 24 | CAM050 | Kimberley | 14.85 | 123.8 | 2 | 65 | 5 | 58 | 58 | -35 | -38 | 7 | 7 | 64 | 61 | -3 | 17 | 18 | -5 | 29 | 34 |
| 25 | Darwin_CW3 | Arnhem | 11.95 | 131.23 | 9 | 98 | 3 | 22 | 20 | 46 | 49 | 7 | 3 | 76 | 88 | 12 | 15 | 24 | 15 | 32 | 35 |
| 26 | ARA-GOC87 | GOC | 9.818 | 137.12 | 3 | 61 | 2 | 47 | 46 | 81 | 79 | 7 | 6 | 21 | 24 | 3 | -14 | 6 | 13 | 30 | 36 |
| 27 | Darwin_CW2 | Arnhem | 12.06 | 130.95 | 8 | 61 | 4 | 34 | 30 | 73 | 75 | 0 | 4 | 83 | 97 | 14 | 15 | 28 | 17 | 34 | 36 |
| 28 | ITFJBG | Arnhem | 13.61 | 128.97 | 1 | 226 | 4 | 61 | 56 | -29 | -31 | -10 | -18 | 34 | 44 | 10 | 6 | 10 | 28 | 31 | 39 |
| 29 | Cape Capricorn | SGBR | 23.51 | 151.29 | 2 | -56 | 9 | 26 | 27 | -37 | -37 | -7 | -7 | 39 | 29 | -9 | 2 | 9 | -24 | 24 | 39 |
| 30 | CAM100 | Kimberley | 14.32 | 123.6 | 5 | 92 | 12 | 99 | 96 | -37 | -39 | 9 | 9 | 47 | 49 | 2 | 14 | 12 | 5 | 25 | 40 |
| 31 | GBRCCH | SGBR | 22.41 | 151.99 | 6 | 123 | 7 | 93 | 87 | -70 | -68 | 0 | -1 | 28 | 33 | 5 | 8 | 7 | 18 | 24 | 40 |
| 32 | CW6A1 | Bass | 40.43 | 148.54 | 16 | 35 | 9 | 37 | 33 | 22 | 24 | 7 | 9 | 36 | 27 | -9 | -5 | 10 | -26 | 27 | 42 |
| | mean abs. value | N=32 | | | | | 6 | 55 | 51 | | | | | 64 | 66 | 7 | 8 | 14 | 11 | 21 | 29 |
| 33 | BASS-CS91 | Bass | 38.91 | 143.54 | 2 | 81 | 8 | 64 | 56 | 67 | 81 | 4 | 3 | 38 | 49 | 11 | -6 | 12 | 30 | 32 | 44 |
| 34 | NW Shelf M6 | Pilbara | 19.74 | 116.39 | 3 | 112 | 6 | 65 | 64 | -54 | -44 | 5 | 4 | 25 | 33 | 7 | 6 | 8 | 30 | 31 | 44 |
| 35 | GBROTE | SGBR | 23.48 | 152.17 | 4 | -22 | 17 | 60 | 61 | 70 | 72 | 10 | 4 | 29 | 31 | 3 | -3 | 3 | 9 | 11 | 45 |
| 36 | TIMORS88 | Kimberley | 12.76 | 125.66 | 2 | 23 | 4 | 91 | 92 | 47 | 54 | -1 | 7 | 23 | 31 | 8 | -5 | 9 | 36 | 37 | 46 |
| 37 | Round Hill Hd | SGBR | 24.11 | 151.96 | 1 | 218 | 9 | 26 | 25 | -75 | -75 | 0 | 0 | 16 | 14 | -2 | -3 | 2 | -14 | 15 | 47 |
| 38 | Tas91UNSW1_65m | Bass | 40.84 | 144.14 | 3 | 111 | 4 | 95 | 93 | 27 | 35 | 6 | 8 | 15 | 19 | 5 | -1 | 5 | 31 | 31 | 47 |
| 39 | BASS-UN91 | Bass | 41.18 | 144.23 | 5 | 151 | 6 | 115 | 116 | 42 | 32 | 8 | 4 | 14 | 11 | -3 | -7 | 4 | -22 | 25 | 48 |
| 40 | BASS-CS91 | Bass | 38.5 | 148 | 3 | -58 | 11 | 70 | 65 | 72 | 69 | 8 | 10 | 24 | 29 | 6 | 2 | 6 | 24 | 25 | 48 |
| 41 | SAM6IS | SA | 35.5 | 136.6 | 3 | 188 | 8 | 83 | 85 | 57 | 55 | 0 | 0 | 9 | 9 | -1 | 0 | 1 | -7 | 7 | 49 |
| 42 | Darwin_C1 | Arnhem | 12.13 | 131.05 | 6 | 51 | 4 | 52 | 30 | -102 | -89 | -1 | 1 | 156 | 83 | -73 | 11 | 76 | -47 | 49 | 50 |
| 43 | PIL050 | Pilbara | 20.05 | 116.42 | 2 | 268 | 12 | 55 | 52 | -50 | -49 | 4 | 2 | 25 | 31 | 6 | 5 | 6 | 24 | 26 | 51 |
| 44 | CW6A2 | Bass | 40.43 | 148.53 | 4 | 94 | 6 | 31 | 33 | 46 | 24 | 4 | 9 | 45 | 27 | -19 | 12 | 20 | -41 | 44 | 51 |
| 45 | PIL100 | Pilbara | 19.69 | 116.11 | 7 | 223 | 13 | 105 | 114 | -53 | -51 | 2 | 2 | 21 | 25 | 4 | 4 | 4 | 19 | 20 | 51 |
| 46 | ITFTIS | Arnhem | 9.818 | 127.55 | 2 | 223 | 7 | 464 | 534 | -97 | -86 | 1 | 2 | 8 | 8 | 0 | -2 | 1 | 6 | 7 | 51 |
| 47 | Wigton I | SGBR | 20.67 | 149.47 | 6 | 66 | 9 | 38 | 39 | -8 | -2 | 3 | 6 | 40 | 44 | 5 | 22 | 17 | 12 | 42 | 53 |
| 48 | PIL200 | Pilbara | 19.44 | 115.92 | 8 | 231 | 11 | 208 | 239 | -73 | -67 | 0 | 0 | 13 | 15 | 2 | -2 | 2 | 14 | 15 | 55 |
| 49 | NRSYON | CGBR | 19.3 | 147.62 | 1 | -30 | 18 | 30 | 29 | -29 | -34 | 9 | 11 | 18 | 16 | -2 | 5 | 2 | -11 | 14 | 57 |
| 50 | Darwin_CW1 | Arnhem | 12.1 | 131.12 | 7 | 199 | 4 | 22 | 21 | -90 | -85 | 0 | 7 | 108 | 46 | -63 | 4 | 63 | -58 | 58 | 59 |
| 51 | KIM400 | Kimberley | 15.22 | 121.11 | 1 | -85 | 7 | 396 | 371 | -64 | -60 | 5 | 5 | 10 | 13 | 3 | -6 | 3 | 33 | 35 | 62 |
| 52 | ARA-GOC87 | GOC | 13.99 | 139.03 | 2 | -2 | 3 | 60 | 62 | -54 | -64 | 4 | 3 | 7 | 8 | 2 | 20 | 3 | 28 | 48 | 63 |
| 53 | BASS-CS91 | Bass | 38 | 148 | 1 | 137 | 10 | 47 | 45 | 72 | 66 | 2 | 2 | 12 | 15 | 3 | -9 | 4 | 27 | 32 | 64 |
| 54 | ITFMHB | Arnhem | 11 | 128 | 1 | 74 | 9 | 146 | 130 | -34 | -41 | -6 | -4 | 14 | 18 | 5 | -15 | 6 | 33 | 44 | 66 |
| 55 | SAM8SG | SA | 35.25 | 136.69 | 2 | 92 | 10 | 53 | 61 | 42 | 35 | 3 | 2 | 9 | 11 | 2 | -11 | 3 | 20 | 28 | 67 |
| 56 | GBRPPS | CGBR | 18.31 | 147.17 | 5 | 205 | 15 | 72 | 71 | 40 | 32 | 4 | 4 | 13 | 17 | 4 | 0 | 4 | 32 | 32 | 69 |
| 57 | Brampton I | SGBR | 20.85 | 149.27 | 2 | 5 | 10 | 18 | 18 | -9 | -5 | -4 | 4 | 32 | 43 | 11 | 27 | 20 | 33 | 62 | 72 |





| Row | Site | Region | lat. | long. | \|mean\| | dir | sub_o | h_o | h_m | inc_o | inc_m | min_o | min_m | maj_o | maj_m | m-o | m-o | \|m̄-ō\| | reM2 | reM2̄ | reLF |
|---|---|---|---|---|---|---|---|---|---|---|---|---|---|---|---|---|---|---|---|---|---|
| | | | °S | °E | cm s⁻¹ | °T | cm s⁻¹ | m | m | °T | °T | cm s⁻¹ | cm s⁻¹ | cm s⁻¹ | cm s⁻¹ | cm s⁻¹ | Δ° | cm s⁻¹ | % | % | % |
| 58 | NRSKAI | SA | 35.83 | 136.45 | 12 | 192 | 20 | 103 | 110 | 17 | 13 | 0 | 0 | 8 | 7 | -1 | -8 | 1 | -10 | 17 | 76 |
| 59 | TASE88 | Tas | 42.65 | 148.28 | 9 | 5 | 13 | 110 | 104 | -2 | -1 | 1 | 0 | 6 | 4 | -1 | -5 | 1 | -25 | 26 | 77 |
| 60 | SAM2CP | SA | 35.28 | 135.67 | 5 | -36 | 13 | 100 | 99 | 56 | 52 | 1 | 0 | 4 | 5 | 0 | -2 | 0 | 10 | 10 | 77 |
| 61 | GBRLSH | CGBR | 14.7 | 145.63 | 2 | -84 | 15 | 32 | 31 | 15 | 70 | 2 | 1 | 13 | 9 | -3 | -32 | 7 | -26 | 54 | 79 |
| 62 | NRSMAI | Tas | 42.6 | 148.23 | 5 | 18 | 15 | 90 | 93 | -4 | -9 | 1 | 0 | 6 | 4 | -2 | 9 | 2 | -27 | 31 | 81 |
| 63 | N Bugatti Rf | SGBR | 20.03 | 150.3 | 12 | 54 | 7 | 64 | 47 | 19 | 45 | 8 | 6 | 48 | 86 | 38 | 5 | 39 | 80 | 81 | 83 |
| 64 | W Bugatti Rf | SGBR | 20.08 | 150.25 | 13 | 178 | 3 | 70 | 51 | 33 | 11 | 4 | 20 | 55 | 99 | 44 | 10 | 46 | 80 | 83 | 84 |
| | mean abs. value | N=32 | | | | | 10 | 95 | 95 | | | | | 27 | 27 | 11 | 8 | 12 | 39 | 44 | 59 |
| 65 | Creal Rf | SGBR | 20.5 | 150.4 | 3 | 230 | 3 | 69 | 69 | 17 | 16 | 8 | 9 | 23 | 39 | 15 | 24 | 20 | 66 | 85 | 87 |
| 66 | GBRELR | SGBR | 21.04 | 152.89 | 48 | 116 | 41 | 305 | 316 | 58 | 78 | 1 | 0 | 5 | 6 | 1 | -1 | 1 | 18 | 18 | 91 |
| 67 | SAM5CB | SA | 34.93 | 135.01 | 2 | 104 | 23 | 98 | 95 | 14 | 12 | 1 | 1 | 3 | 3 | 1 | 0 | 1 | 26 | 26 | 93 |
| 68 | SAM3MS | SA | 36.15 | 135.9 | 18 | 142 | 21 | 168 | 160 | 55 | 30 | 2 | 1 | 3 | 3 | 0 | -25 | 1 | 9 | 46 | 94 |
| 69 | CH100 | SEQ | 30.26 | 153.4 | 31 | 199 | 37 | 97 | 92 | -14 | -67 | 1 | 1 | 2 | 2 | 0 | 10 | 0 | 3 | 17 | 95 |
| 70 | CH070 | SEQ | 30.27 | 153.3 | 18 | 200 | 27 | 76 | 92 | -17 | -67 | 1 | 1 | 2 | 2 | 0 | 19 | 1 | 19 | 41 | 96 |
| 71 | BMP070 | NSW | 36.19 | 150.19 | 10 | 182 | 17 | 74 | 61 | -20 | -28 | 0 | 1 | 1 | 1 | 0 | -21 | 1 | 22 | 45 | 96 |
| 72 | WATR04 | SW | 31.72 | 115.4 | 2 | -56 | 18 | 46 | 42 | 66 | 59 | 0 | 0 | 0 | 0 | 0 | 1 | 0 | 10 | 11 | 98 |
| 73 | BMP120 | NSW | 36.21 | 150.32 | 14 | 173 | 35 | 121 | 125 | -29 | -45 | 0 | 1 | 1 | 1 | 0 | -6 | 0 | 35 | 37 | 98 |
| 74 | SAM7DS | SA | 36.2 | 135.84 | 7 | 150 | 11 | 519 | 587 | 55 | 30 | 1 | 0 | 1 | 2 | 1 | -36 | 1 | 41 | 84 | 98 |
| 75 | SYD140 | NSW | 34 | 151.45 | 16 | 205 | 27 | 138 | 144 | 10 | -20 | 1 | 1 | 2 | 2 | 0 | 37 | 1 | 22 | 73 | 99 |
| 76 | SYD100 | NSW | 33.94 | 151.38 | 14 | 199 | 26 | 103 | 117 | 5 | -17 | 1 | 1 | 2 | 2 | 1 | 35 | 1 | 36 | 79 | 99 |
| 77 | NRSROT | SW | 32 | 115.42 | 1 | 180 | 32 | 47 | 42 | 59 | 81 | 0 | 0 | 0 | 1 | 0 | -2 | 0 | 38 | 38 | 99 |
| 78 | WACA20 | SW | 31.98 | 115.23 | 9 | 168 | 20 | 199 | 212 | 42 | 87 | 0 | 0 | 0 | 0 | 0 | 43 | 0 | 14 | 80 | 100 |
| 79 | PH100 | NSW | 34.12 | 151.23 | 7 | 224 | 21 | 110 | 123 | 29 | -7 | 1 | 1 | 1 | 1 | 0 | 54 | 1 | 23 | 104 | 100 |
| 80 | WATR20 | SW | 31.73 | 115.04 | 16 | 169 | 28 | 205 | 167 | 11 | 53 | 0 | 0 | 0 | 0 | 0 | 70 | 0 | 31 | 135 | 100 |
| 81 | WATR50 | SW | 31.76 | 114.96 | 6 | 170 | 15 | 497 | 469 | -5 | 32 | 0 | 0 | 0 | 0 | 0 | 71 | 0 | 20 | 129 | 100 |
| 82 | WATR15 | SW | 31.69 | 115.13 | 10 | 165 | 26 | 150 | 160 | 178 | 48 | 0 | 0 | 0 | 0 | 0 | -83 | 0 | 23 | 149 | 101 |
| 83 | NRSESP | SW | 33.93 | 121.85 | 1 | 107 | 5 | 50 | 44 | 44 | 59 | 0 | 0 | 0 | 1 | 0 | -17 | 0 | 99 | 108 | 101 |
| 84 | GBRMYR | CGBR | 18.22 | 147.35 | 13 | 113 | 17 | 214 | 190 | 37 | 34 | 2 | 2 | 6 | 12 | 6 | -17 | 6 | 93 | 102 | 101 |
| 85 | SAM4CY | SA | 36.53 | 136.87 | 0 | -30 | 22 | 117 | 105 | 18 | 59 | 0 | 1 | 1 | 2 | 1 | -9 | 1 | 120 | 123 | 101 |
| 86 | SEQ400 | SEQ | 27.33 | 153.88 | 28 | 183 | 39 | 400 | 373 | 49 | 75 | 0 | 1 | 1 | 2 | 0 | 60 | 2 | 33 | 121 | 101 |
| 87 | BMP090 | NSW | 36.19 | 150.23 | 20 | 172 | 19 | 91 | 96 | -154 | -34 | 0 | 0 | 1 | 1 | 1 | 37 | 1 | 93 | 129 | 101 |
| 88 | WATR10 | SW | 31.65 | 115.2 | 9 | 150 | 18 | 107 | 79 | 137 | 53 | 0 | 0 | 0 | 0 | 0 | -82 | 1 | 129 | 236 | 102 |
| 89 | L Musgrave I | SGBR | 23.93 | 152.3 | 3 | 166 | 5 | 42 | 42 | 85 | 63 | 2 | 6 | 14 | 29 | 15 | 0 | 15 | 105 | 105 | 103 |
| 90 | SEQ200 | SEQ | 27.34 | 153.77 | 23 | 178 | 44 | 200 | 203 | 110 | 87 | 0 | 1 | 1 | 3 | 2 | -75 | 3 | 238 | 326 | 104 |
| 91 | SAM1DS | SA | 36.52 | 136.24 | 5 | 114 | 10 | 520 | 587 | -14 | 25 | 0 | 0 | 0 | 1 | 1 | 22 | 1 | 340 | 350 | 108 |
| 92 | NRSNSI | SEQ | 27.34 | 153.56 | 25 | 159 | 33 | 65 | 63 | -97 | -84 | 1 | 3 | 3 | 9 | 6 | -15 | 6 | 229 | 233 | 110 |
| 93 | Tern I | SGBR | 20.85 | 149.98 | 8 | 141 | 7 | 47 | 50 | 28 | 4 | 1 | 5 | 22 | 35 | 13 | 57 | 29 | 59 | 133 | 125 |





| Row | Site | Region | lat. | long. | \|mean\| | dir | sub_o | h_o | h_m | inc_o | inc_m | min_o | min_m | maj_o | maj_m | m-o | m-o | \|m̄-ō\| | reM2 | reM̄2 | reLF |
|---|---|---|---|---|---|---|---|---|---|---|---|---|---|---|---|---|---|---|---|---|---|
| | | | °S | °E | cm s⁻¹ | °T | cm s⁻¹ | m | m | °T | °T | cm s⁻¹ | cm s⁻¹ | cm s⁻¹ | cm s⁻¹ | cm s⁻¹ | Δ° | cm s⁻¹ | % | % | % |
| 94 | NRSNIN | Pilbara | 21.87 | 113.95 | 4 | 211 | 18 | 61 | 64 | -131 | -77 | 0 | -4 | 7 | 20 | 14 | 7 | 14 | 208 | 209 | 129 |
| 95 | GBRLSL | CGBR | 14.34 | 145.34 | 4 | -61 | 12 | 330 | 480 | 10 | 39 | 0 | 0 | 0 | 7 | 7 | -3 | 7 | 2478 | 2478 | 157 |
| | mean abs. value | N=31 | | | | | 22 | 170 | 176 | | | | | 3 | 6 | 3 | 30 | 4 | 83 | 112 | 102 |
| | mean abs. value | N=95 | | | | | 13 | 106 | 107 | | | | | 32 | 33 | 7 | 15 | 10 | 22 | 31 | 51 |


**Table 4: Tidal major axis velocity and phase region-average statistics, for eight constituents (and their root sum of squares).**

**mean observed major axis amplitude < o >  (cm s⁻¹)**

| | Aust | Arnhem | GOC | CGBR | SGBR | SEQ | NSW | Bass | Tas | SA | SW | Pilbara | Kimb. |
|---|---|---|---|---|---|---|---|---|---|---|---|---|---|
| #sites | 95 | 12 | 3 | 5 | 15 | 5 | 6 | 18 | 10 | 9 | 8 | 5 | 7 |
| M2 | 31.7 | 69.5 | 16.3 | 9.9 | 31.8 | 1.8 | 1.2 | 66.3 | 64.1 | 4.3 | 0.33 | 18 | 35.5 |
| S2 | 11.3 | 32.8 | 5.3 | 5.3 | 12.9 | 0.62 | 0.41 | 10.2 | 9.1 | 4.6 | 0.4 | 11 | 21.7 |
| N2 | 6.1 | 10.9 | 3.6 | 3.2 | 7.4 | 0.51 | 0.37 | 13.7 | 13.3 | 0.41 | 0.15 | 3 | 5.9 |
| K2 | 2.9 | 7.3 | - | 1.3 | 3.7 | 0.25 | 0.12 | - | 0.25 | 1.2 | 0.12 | 2.8 | 6.6 |
| K1 | 6.7 | 16.8 | 17.4 | 2.8 | 5.3 | 3.3 | 3.1 | 7.7 | 7.1 | 5.6 | 0.74 | 2.7 | 4.4 |
| O1 | 4 | 9.5 | 9.7 | 1.5 | 3 | 3.1 | 2 | 5.3 | 5 | 3.5 | 0.54 | 1.4 | 2.2 |
| P1 | 1.6 | 3.5 | - | 1.1 | 1.9 | 1.1 | 1.2 | - | 2.6 | 1.8 | 0.49 | 0.78 | 1.2 |
| Q1 | 0.9 | 2 | 2.5 | 0.35 | 0.62 | 0.68 | 0.47 | 1.1 | 1.2 | 0.79 | 0.23 | 0.54 | 0.56 |
| RSS | 35.3 | 80.4 | 26.7 | 12.3 | 35.9 | 5.1 | 4.1 | 69.1 | 66.7 | 9.4 | 1.2 | 21.8 | 42.9 |

**mean magnitude of vector error (MMVE) (cm s⁻¹)**

| | Aust | Arnhem | GOC | CGBR | SGBR | SEQ | NSW | Bass | Tas | SA | SW | Pilbara | Kimb. |
|---|---|---|---|---|---|---|---|---|---|---|---|---|---|
| #sites | 95 | 12 | 3 | 5 | 15 | 5 | 6 | 18 | 10 | 9 | 8 | 5 | 7 |
| M2 | 9.8 | 23.4 | 4.8 | 5.3 | 15 | 2.4 | 0.88 | 14.8 | 14.4 | 1.1 | 0.31 | 6.8 | 7.9 |
| S2 | 4 | 12.2 | 3.1 | 3.1 | 5.3 | 1 | 0.37 | 2.9 | 2.6 | 1.3 | 0.33 | 3.3 | 5.2 |
| N2 | 2.1 | 3.9 | 1.4 | 1.7 | 3.6 | 0.57 | 0.24 | 3.4 | 3.1 | 0.15 | 0 | 1.3 | 1.3 |
| K2 | 0.96 | 1.9 | - | 0.76 | 1.5 | 0.26 | 0 | - | 0.15 | 0.46 | 0.12 | 0.88 | 1.7 |
| K1 | 3.2 | 7 | 6.2 | 1.6 | 2.5 | 2.8 | 2.5 | 3.6 | 3.7 | 2.6 | 0.67 | 1.1 | 1.5 |
| O1 | 2.2 | 4.7 | 5.3 | 0.79 | 1.5 | 2.7 | 1.8 | 2.5 | 2.7 | 1.7 | 0.4 | 0.58 | 1.1 |
| P1 | 0.86 | 1.4 | - | 0.58 | 1 | 0.96 | 1 | - | 2.3 | 0.87 | 0.4 | 0.36 | 0.41 |
| O1 | 0.49 | 0.91 | 0.47 | 0.21 | 0.42 | 0.59 | 0.46 | 0.6 | 0.82 | 0.41 | 0.16 | 0.36 | 0.19 |
| RSS | 11.6 | 28.1 | 10 | 6.7 | 16.6 | 4.8 | 3.4 | 16.1 | 15.8 | 3.7 | 1 | 7.9 | 9.9 |
| %obs | 32.8 | 34.9 | 37.7 | 54.8 | 46.4 | 94.7 | 83.3 | 23.3 | 23.7 | 39.6 | 84.3 | 36.2 | 23 |



**mean absolute value of error <|m-o|> (MAE) (cm s$^{-1}$)**

|        | Aust | Arnhem | GOC  | CGBR | SGBR | SEQ  | NSW  | Bass | Tas  | SA   | SW   | Pilbara | Kimb. |
|--------|------|--------|------|------|------|------|------|------|------|------|------|---------|-------|
| #sites | 95   | 12     | 3    | 5    | 15   | 5    | 6    | 18   | 10   | 9    | 8    | 5       | 7     |
| M2     | 6.9  | 18.3   | 2.8  | 4.4  | 11.4 | 1.8  | 0.4  | 8.7  | 7.8  | 0.79 | 0.14 | 6.6     | 3.4   |
| S2     | 2.9  | 10.1   | 1.2  | 2.4  | 3.7  | 0.72 | 0.18 | 2.1  | 1.8  | 1.1  | 0.15 | 3.1     | 1.7   |
| N2     | 1.4  | 2.6    | 0.36 | 1.5  | 2.7  | 0.34 | 0    | 2.2  | 2    | 0    | 0    | 1.2     | 0.55  |
| K2     | 0.6  | 1.3    | -    | 0.54 | 0.96 | 0.21 | 0    | -    | 0.11 | 0.34 | 0    | 0.84    | 0.5   |
| K1     | 2.5  | 5.3    | 3.2  | 1.2  | 1.9  | 2.2  | 2.4  | 3    | 3    | 2.3  | 0.2  | 0.61    | 1     |
| O1     | 1.7  | 3.3    | 4    | 0.65 | 0.95 | 2.2  | 1.6  | 2    | 2    | 1.4  | 0.25 | 0.5     | 0.87  |
| P1     | 0.6  | 0.75   | -    | 0.37 | 0.66 | 0.78 | 0.99 | -    | 2.1  | 0.79 | 0.24 | 0.3     | 0.15  |
| Q1     | 0.34 | 0.62   | 0.43 | 0.16 | 0.24 | 0.51 | 0.38 | 0.35 | 0.57 | 0.31 | 0.13 | 0.26    | 0     |
| RSS    | 8.2  | 22     | 6    | 5.5  | 12.6 | 3.9  | 3.1  | 9.9  | 9.3  | 3.1  | 0.48 | 7.4     | 4.1   |
| %obs   | 23.3 | 27.4   | 22.5 | 44.6 | 35   | 75.6 | 75   | 14.3 | 14   | 33.3 | 39.9 | 34.2    | 9.7   |

**mean error < m-o > (bias) (cm s$^{-1}$)**

|        | Aust  | Arnhem | GOC  | CGBR | SGBR  | SEQ   | NSW   | Bass  | Tas   | SA    | SW    | Pilbara | Kimb. |
|--------|-------|--------|------|------|-------|-------|-------|-------|-------|-------|-------|---------|-------|
| #sites | 95    | 12     | 3    | 5    | 15    | 5     | 6     | 18    | 10    | 9     | 8     | 5       | 7     |
| M2     | 1.7   | -7.4   | 2.8  | 2.3  | 9.5   | 1.8   | 0.4   | 1.3   | 0.76  | 0.45  | 0.14  | 6.6     | 2.4   |
| S2     | 0.23  | -2.4   | 1.2  | 1.3  | 2.9   | 0.54  | 0.12  | -1.8  | -1.6  | 0.85  | 0     | 3.1     | 0.53  |
| N2     | 0.13  | -1.4   | 0.36 | 0.37 | 2.1   | 0.24  | 0     | -0.65 | -0.94 | 0     | 0     | 1.2     | 0     |
| K2     | 0.27  | 0.46   | -    | 0.39 | 0.42  | 0     | 0     | -     | 0.11  | 0.34  | 0     | 0.84    | -0.22 |
| K1     | -0.86 | -0.87  | -3.2 | 0.46 | 1.1   | -2    | -2.4  | -2.1  | -2.8  | -1.6  | 0     | 0.59    | 0.94  |
| O1     | -0.52 | -0.44  | 3.1  | 0    | -0.12 | -2.2  | -1.6  | -1.4  | -1.7  | -0.76 | 0     | 0.5     | 0.87  |
| P1     | -0.25 | 0.16   | -    | 0    | 0     | -0.76 | -0.99 | -     | -2.1  | -0.68 | -0.24 | 0.17    | 0.15  |
| Q1     | -0.19 | -0.23  | 0.2  | 0    | -0.16 | -0.51 | -0.38 | -0.22 | -0.47 | -0.16 | -0.1  | -0.13   | 0     |





**Figure 6 Amplitude of the M2 major axis velocity for Bass Strait, otherwise like Fig. 5, except that percentiles of the model at the locations of the observations are not listed.**




**Figure 7 Amplitude of the M2 major axis velocity for Banks Strait, otherwise like Fig. 5.**



### 6.2.1 Bass Strait (including Banks Strait)

The tide comes into Bass Strait from both the east and west, with the strongest flows (Fig. 6) either side of the central basin (see Fig. 2) where the tidal range (Fig. 3) is a maximum. The highest tidal ranges are near Burnie on the northern Tasmanian
coast. Recalling that tidal potential forcing is not activated in this run of the model, the agreement of our model with the observations is in contrast with the conclusion by Wijeratne et al. (2012) that tidal potential forcing is required for a nested model of Bass Strait to be accurate. We offer no explanation of this inconsistency. The greatest observed M2 major axis amplitude is 144 cm s$^{-1}$ (at C1A3 in Banks Strait – see Fig. 7, one of the Penesis et al. (2020) ADCPs), where the model estimate is 120 cm s$^{-1}$ (line 16 of Table 3). This is also the biggest error in Bass Strait, but it is still quite a small (-17%) relative
error of amplitude. Taking the phase error also into account takes this to 26%. Table 4 lists the M2 MAE across the 18 validation sites in Bass Strait as 8.7 cm s$^{-1}$. The RSS across 8 constituents is 9.9 cm s$^{-1}$, or 14.3 % of the 69 cm s$^{-1}$ mean observed RSS of amplitudes – a much better than average (23% across Australia) relative error. Figure 6 and Table 3 show that, across Bass Strait, the modelled M2 current ellipse eccentricities and orientations are mostly in good agreement with observations. The phase errors range from -9° to 12°. Summing over 8 constituents, and taking the phase errors into account
(Table 4), the RSS MMVE is 16.1 cm s$^{-1}$, or 23.3 % of the mean observed RSS amplitude, making Bass Strait the region with the equal lowest (with the Kimberley) relative error of RSS MMVE. See below for a discussion of the M4 constituent.





**Figure 8 Amplitude of the M2 major axis velocity for the Kimberley, otherwise like Fig. 5.**

## 6.2.2 Kimberley

The Kimberley region of Australia includes King Sound, where the greatest tidal range in Australia occurs. The entrance to King Sound has such strong tidal currents that tourists go out to see them in RIBs, helicopters and other vessels. There are not, however, any available instrumental records of the flows in the most energetic regions, so the percentiles of the model (across ~30,000 cells, see Fig. 8) are very different to the percentiles of the observations. Figure 8 shows that the model agrees quite well with the seven available records, including the change from nearly circular M2 ellipses at KIM050 to the shore-normal



rectilinear flows at CAM050 and CAM100, and then the weak shore-parallel ellipses at TIMORS88. The M2 amplitude errors
      at KIM100 and KIM200 are just 3 and -1 % of the observed amplitude. It is only with the phase taken into account that the
      M2 relative errors are significant (11 and 1%). The RSS MMVE is 9.9 cm s⁻¹, or 23 % of the observed RSS amplitude, like
      the Bass Strait figure.



**Figure 9 Amplitude of the M2 major axis velocity for the Darwin region, otherwise like Fig. 5.**



**Figure 10 Amplitude of the M2 height for the Darwin region, otherwise like Fig. 3.**

### 6.2.3 Darwin

Figure 9 shows that M2 velocity errors are relatively low at six of the eight sites in the Darwin–Clarence Channel region. Table

3 (lines 42 and 50) identifies the two noticeable exceptions as being the Darwin-C1 and CW1 sites, where the M2 major axis

amplitude errors are -73 and -63 cm s$^{-1}$. At C1 the problem is clearly the topography; model depth is only 30 m but the in situ

depth is 52 m. It is less clear why the error at CW1 is large but we will not be surprised if rebuilding the mesh using recently-

acquired topography data does not reduce these errors. At present however, the velocity major axis RSS MMVE for Arnhem

remains listed as 28.1 cm s$^{-1}$, or 34.9 % of the observed RSS amplitude. The modelled tidal height amplitude in Van Diemen

Gulf (Fig. 10, see Christine Reef for example) is significantly weaker than the observations, for reasons that we are yet to

determine.





**Figure 11 Amplitude of the M2 major axis velocity for the Southern Great Barrier Reef region, otherwise like Fig. 5.**

### 6.2.4 Southern Great Barrier Reef

The Barrier Reef is dense off Broad Sound, causing tides to enter the reef lagoon from both the NW and SE. These waves meet

in the lagoon outside Broad Sound then further amplification of the wave entering the Sound occurs due to the geometry of

the Sound (Middleton, Buchwald and Huthnance, 1984). Our model simulates the first process satisfactorily in a qualitative



sense (see Fig. 11), and the modelled and observed tidal currents are in very good agreement at many locations. But Table 4 also lists some large discrepancies at several sites. These are where the observations were made by mechanical current meters,

some in topographically complex locations (two near Bugatti Reef, one near Lady Musgrave Island), so the listed RSS MMVE of 16.6 cm s$^{-1}$ (or 46.4 % of the observed amplitude) possibly overstates the true error. The tide gauge (at McEwin Islet)  near the head of the Sound (Fig. 3) suggests that the second amplification process is not well modelled, since the modelled range there is only about 75% of the observed range, and the modelled tide lags the observed tide by about 2 h.




**Figure 12 Amplitude of the M2 major axis velocity for the South Australia region, otherwise like Fig. 5.**

### 6.2.5 South Australia

A distinctive feature of the tides of South Australia is that the amplitude of S2 exceeds that of M2 (barely), leading to a very
strong spring-neap cycle. The vanishing semidiurnal tide on days when M2 and S2 are out of phase is locally known as the
Dodge tide. Table 2 lists the SA-average observed M2 and S2 height and major axis amplitudes as 25.5 and 26.7 cm, and 4.3
and 4.6 cm s$^{-1}$. The model M2 and S2 height and major axis amplitudes (not listed) are also nearly equal, at 23 and 27 cm, and
4.7 and 5.4 cm s$^{-1}$ so Dodge tides will also occur (imperfectly) in model-generated predictions. The maximum modelled M2





major axis amplitude is 41 cm s$^{-1}$ in the South Australian region (Fig. 12), but we have no observations to validate the model
at that location. The maximum observed M2 major axis amplitude is 9 cm s$^{-1}$ at both SAM6IS and SAM8SG (rows 41 and 55
of Table 3) where the model is in very close and good agreement, respectively. The RSS MMVE for SA is 3.7 cm s$^{-1}$, or 39.6 %
of the observed amplitude.

**6.2.6 Pilbara**

Table 3 lists results for just five sites in the Pilbara region (one being the Ningaloo site mentioned earlier as having the greatest
error). Unfortunately, these are all we have in our validation dataset despite the economic importance of marine traffic in this
region. Results for the three IMOS ADCPs near 20° S (PIL050, 100 and 200) include M2 vector errors of 15 to 26 % of the
observed amplitude. But this region is well known for strong internal tides (Book et al., 2016), to which our analysis method
is essentially blind, and thus underestimates the errors. Internal tides aside, the RSS MMVE for this region is 7.9 cm s$^{-1}$, or
36.2 % of the observed amplitude.

**6.2.7 Gulf of Carpentaria, Torres Strait, Central Great Barrier Reef**

The GOC and CGBR regions have intermediate (37.7 and 54.8 %) relative errors of the RSS MMVE, but being based on just
3 and 5 sites, these statistics are uncertain. Nevertheless, we see value in publishing tidal current predictions for these two
regions, with appropriate warnings, partly because the sub-tidal currents are weak in these two regions. As mentioned earlier,
Torres Strait is one of the few places where official tidal current predictions are already published. We have not yet compared
those predictions, or observation-based constituents with our model.

**6.2.8 South-east Queensland, New South Wales and South West**

The relative error of the RSS MMVE for the SEQ, NSW and SW regions are 95, 83 and 84 %, respectively, suggesting that
the model is not simulating the tidal currents in these regions very well, even though it is simulating the heights (recall that
NSW is one of the regions with the lowest relative error of height). These narrow-shelf regions are also where the sub-tidal
currents (Table 3) far exceed the tidal currents, so predictions of tidal currents would be of limited practical value even if they
were accurate. For both these reasons, we will not be publishing tidal current predictions from the COMPAS model for these
regions.



**Figure 13 Amplitude of the M4 major axis velocity for the Banks Strait region, otherwise like Fig. 5.**



### 6.2.9 High frequency constituents

As mentioned in sections 2 and 3, we have analysed both the model and the velocity validation data set for 13 tidal constituents. Table 4 does not include results for M4, MS4, M6, 2MS6 or 2N2 because the amplitudes are mostly insignificant. An exception is the M4 constituent in Banks Strait, where 5.9 cm s$^{-1}$ was observed (Fig. 13). Model amplitudes are comparable but the
inclinations and phases are not accurate enough to warrant inclusion of these constituents when making predictions.

### 7 Discussion

We have evaluated the tidal heights in our COMPAS model against a large number (615) of sites around Australia, giving a much more detailed picture than was given, for example, by Haigh et al. (2014) or Seifi et al. (2019), while being broadly
consistent. But modelling tidal heights is not the principal motivation of this study. Our focus is on tidal currents (depth-averaged at this point), about which much less has been written (Stammer et al., 2014; Timko et al., 2013). Lyard et al. (2020) compare FES2014 with the IMOS component of the validation data we have used (just graphically). They conclude that for shelf currents, there is still a need for nested regional models (such as ours), with finer grids than global models have.

We have shown that our COMPAS model of the barotropic tide is in very good agreement with observed tidal currents at
many, but certainly not all, of the 95 sites at which we have in situ validation data. A large number of the sites with high relative errors are where the tides are very weak, so it could be argued that those errors are of little practical interest. Over the continental shelf, this is the case for the southern half of the continent from Ningaloo Reef in the west to Fraser Island in the east, excepting Bass Strait and the South Australian gulfs (i.e. the sections where the shelf is narrow). This leaves 79 % by area of Australia's shelf waters as being where tidal currents are both predictable and a significant proportion of the total
variance.  Bass Strait and the Kimberley region are where our model performs best, with the root sum (across 8 constituents) squared, regional-average vector error of the major axis velocity being 23% of the observed signal. This measure of the relative error of the model's tidal predictions is between 35 and 55 % in the other regions where we think the predictions should be made available to the public.

We hope to expand our tidal currents validation dataset, especially at locations (mainly in the NW) where observations have
been made by offshore industries, in order to guide development of the next version of our model. Incomplete as it is, we are publishing it now because we are sure it will have enduring value, for example, to developers of global models such as Lyard et al (2020) who used a preliminary version of the validation dataset as noted above.

It is well established (e.g. by Ray et al., 2011) that accurate topography is an essential component of a good tidal model and our results and those of Sahuc et al. (2020) bear this out. Some of the largest model errors are where there is a big discrepancy
between the depth in the model and the depth that was recorded on site during mooring deployment. Improving the topography in our model is certainly a priority for future model development. This will likely comprise a combination of inverse tuning





where local bathymetry alterations are made to optimally correlate model predictions to observation, and capitalising on the results of the ausSeabed initiative (http://www.ausseabed.gov.au/about).

Boundary conditions are also, of course, an essential input for a regional tidal model. We have only tested our model using open boundary forcing from one of the several available global models (TPXO9v1). On advice from the model developers, we nested within the 1/6° model rather than the 1/30° 'atlas' (composite) product. The question naturally arises whether our model out-performs the atlas product. At the time of writing, the latest version of this is v4. Using the validation data set discussed here (605 of the 615 tide gauge sites, but all 95 current meter sites, to be precise), we have compared the atlas height and velocity errors (for all 8 height constituents and 13 velocity constituents) with the errors of our model. In summary, we find that the atlas errors for height are significantly less than ours (e.g. 10cm vs 18cm for M2 MMVE), but much more for velocity (20 cm s$^{-1}$ vs 10 cm s$^{-1}$ for M2 MMVE). We assume that the lower height errors are a consequence of the fact that many of the tide gauge data are assimilated, while the greater velocity errors may have several causes, such as 1) the simpler grid, 2) bathymetry errors and 3) spurious height gradients resulting from the assimilation of data that is not perfectly dynamically consistent with the model grid.

## 8 Conclusions

We have shown that for many regions around Australia's continental shelf, our model can predict depth-averaged tidal currents with enough accuracy to arguably be operationally useful for mariners and maritime industries. Regions where tidal currents are most predictable and in excess of non-tidal currents include Bass Strait, the Kimberley, Joseph Bonaparte Gulf to Arnhem Land and the southern Great Barrier Reef. Consequently, these are the regions for which we intend to commence publishing 'unofficial' predictions of tidal currents (both model-based and observation-based). They are also the regions of greatest interest to the renewable energy sector, for whom we have published maps based on the model discussed here. We intend also to publish tidal current predictions for the South Australian gulfs, the Pilbara, Gulf of Carpentaria, Torres Strait and the central and northern Great Barrier Reef regions but with a warning that there may be greater errors in these regions. For the rest of Australia (comprising the narrow-shelf regions of the southern half of the continent) we see no need to publish tidal current predictions, largely because the non-tidal currents are dominant.

## 9 Code Availability

COMPAS is supported by CSIRO, Australia and available open source (see CSIRO, 2021). We appreciate the encouragement of the MPAS developers in pursuing this work.



## 10 Data availability

•   Three project data sets have been published by  Herzfeld, et al. (2020)  Herzfeld, Mike; Griffin, David; Hemer, Mark; Rosebrock, Uwe; Rizwi, Farhan; Trenham, Claire (2020): AusTEN National Tidal model data. v3. CSIRO. Data Collection. https://doi.org/10.25919/q8dw-c732:

1.   The first 59 of the 365 days of COMPAS output hourly time series, at all  cell centers, for all state variables

2.   13 harmonic constituents of the COMPAS velocity and height fields, derived from the 365-day model run

3.   13 (11 in places) harmonic constituents of the currents validation dataset, along with subtidal ellipse parameters for 95 locations.

   •   COMPAS-based estimates of Australia's tidal energy resource are also available at

1.   https://nationalmap.gov.au/renewables/

   •   Current meter validation dataset timeseries are available at: https://portal.aodn.org.au/

•   Graphics similar to the Figures in this paper showing results for all 13 constituents, other regions, other variables, and statistical properties of the tidal heights, energy fluxes, etc. http://www.marine.csiro.au/~griffin/ARENA_tides/tides/

## Author contribution

David Griffin assembled the validation data set, performed the model-data comparisons and prepared the manuscript, with contributions from coauthors. Darren Engwirda prepared the model grid. Mike Herzfeld developed and ran the COMPAS

model. Mark Hemer led the main (ARENA) project that this study is part of, contributed to the analyses and maintained linkages with collaborators.

## Competing interests

The authors declare that they have no conflict of interest.

## Disclaimer

The data products of this research are not for navigation. The work is only a step towards an operational product.

## Acknowledgements

This work was performed as part of three projects: 1) the Australian Tidal Energy (AUSTEn) project funded by the Australian Renewable Energy Agency (ARENA) Advancing Renewables Program under agreement number G00902, 2) the CSIRO, Bureau of Meteorology and Royal Australian Navy Bluelink project and 3) the CSIRO Australian National Modelling

Initiative. Tidal constituents for 683 tide gauge sites were kindly provided by the National Operations Centre (NOC) Tidal Unit of the Bureau of Meteorology, who also provided helpful comments on the work as it progressed, and the resulting paper.





Current meter data were provided at 55 sites by Australia's Integrated Marine Observing System (IMOS), which is enabled by the National Collaborative Research Infrastructure Strategy (NCRIS). IMOS is operated by a consortium of institutions as an unincorporated joint venture, with the University of Tasmania as Lead Agent. Topography data were supplied by Geosciences

Australia and the Naval Research Laboratory Digital Bathymetry Data Base. Global tidal constituents from the TPXO model were provided by Lana Erofeeva and Gary Egbert. We also thank several colleagues for comments on the manuscript, including Madeleine Cahill, James Chittleborough, Andy Taylor and Clothilde Langlais.

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





## Figure Captions