# Peer review of "Australian tidal currents – assessment of a barotropic model (COMPAS v1.3.0 rev6631) with an unstructured grid."

_Geoscientific Model Development, 2021_

## Author Response (AR1)

**Foreword**

We thank the two referees and the one community member for their thoughtful and constructive comments on our paper. We have revised our manuscript in response to referee comments as described below and think the paper is now definitely improved, and hope that the Editor invites us to submit it. Our responses to comments are below in red, with new or altered snippets of the revised paper in green.

**Referee 1**

This paper discusses the validation of a new tide model for the waters surrounding Australia. The model is based on a new implementation of shallow water dynamics on an unstructured grid using the EMS modeling system which they have open-sourced. The authors provide a new compilation of tidal current observations in their domain, which should be quite useful for others. They provide nuanced and intelligent discussion of their process of model development (emphasizing details such as the hand-adjustment of topography and implementation of open boundary conditions) which should also help others. They systematically discuss the model-data intercomparison, emphasizing locations where tidal currents are relatively large in comparison with sub-tidal currents, which is appropriate considering the aimed-at operational uses for the model. Overall, the authors have produced a well-organized and thoughtful comparison, with the appropriate level of detail provided, and I think this paper requires only very minor adjustments before publication.
Thank you.

Detailed notes, itemized by line number:

L14: Should this read "Rood Mean Square Error (RMSE)"? Otherwise, why captials?
No, Root Sum Square is correct, because it is over 8 constituents, and we want to know the total error. '(RSS)' could be added, but it is not used again in the abstract.

L15: Two periods.
Oops. Thank you.

Up to L70: This discussion of the grid development will be useful for others. Very good.
Thank you.

L91: Indeed this is unusual, but it is an indication that you have achieved a necessary level of accuracy. Interesting.
Agreed. See below for further discussion of this point.

L100: When I first read this, I did not understand that the tidal synthesis was only used at the preliminary stage of model tuning. Later, at line 155, this is explained. I think this should be explained right away when the tidal synathesis is mentioned.
Sorry, but it seems you have overlooked lines 108-110, which says that the analyses presented in the paper use constituents analysed from a long model run. We have clarified this point by saying "These trial model runs were too short for accurate decomposition into constituents, so we assessed them against…."

L106: Capitalize "TPXO".
Oops. Thank you.

L140: This is a clear explaination of the current meters and ADCP dataset.

Thank you.

L175: Are D and C in the same units, or is C a measure of area? If you believe the model errors are related to this quantity, perhaps it would be better to pliot the error statistics as a function of J. It does not seem that this J is used later, so maybe it can be omitted.
Thank you - this was unclear (and noted by another referee). C has the same units as D. We now say: "where D is the distance (km) to the model grid point, C is the characteristic size (km) of the cell (see Fig. 1),…". Errors are not strongly related to J, and the form of J has little impact on the average error. But if it were omitted, people would ask 'how did you interpolate the model to the obs?'

L182: Pleaase write out the expression for the relative error that includes the sub_o velocity.
Done.

Table 1: Please format the text so that the lower parts of letters are visible. Note, for example, how the "p", "y", and "g" are truncated from several of the place names.
Done

Up to L230: This is a good overview of the errors. Appropriate detail.
Thank you

To L305: A good explanation of why the discussion focusses on only certain stations.
Thank you

L374: It would be useful to label Van Diemen Gulf; although, I guess it is the large body of water enclosing Christine Reef?
Done

Fig 11: I cannot read the place names here. Can you please label Broad Sound?
Done (mentioned current meter sites are in bigger, brighter text and Broad Sound is labelled)

L387: I think I know the location of this gauge, but I don't understand what we are supposed to observe from Fig 3.
We agree that the reference to Fig. 3 was too cryptic, and have inserted an extra Figure here in support of the comment about how the modelled tidal height compares with the observations, which, on closer inspection, is actually better than we had written before. The new text reads "The tide gauge (at McEwin Islet) near the head of the Sound (Fig. 12) suggests that the second amplification process is also quite well modelled, since the modelled M2 amplitude there is nearly (within about 10%) as great as the observed value".

L465: Good to see this basic comparison with TPXO here. You might wish to look at Zaron and Elipot JGR 2021, who compare currents from an earlier version of this model with drifter-derived currents. Alternately, you might find drifter-derived currents are another useful validation dataset.
Thank you. We will consider using drifters and gliders for validating the next version of this model, but first we wish to get access to all the other current meter time series that exist.

L472: I don't have the expertise to comment on whether the model currents are operationally useful. Instead of saying they are "arguably" useful, it would be better if you can describe alernate viewpoints in a more detail. Are there defintions or criteria which would be useful for arguing this question? What criteria should be used to decide if a model is "good enough" to be useful for current predictions vs tidal energy site evaluation?

This is a very hard question and we are quite sure there is not a unique answer - because there are so many potential applications. So we hope that our paper will equip users to assess the adequacy themselves of our tidal model for their application, with us making as few limiting decisions as possible.

**Referee 2**

This paper discusses the comparison of a new tide model for the waters surrounding Australia and both tidal heights and currents observations, with a dedicated focus on future operational tidal currents prediction (from model simulations) added value.

The compilation of tidal observations, especially tidal currents, is rather impressive and will provide a very useful database for further studies and/or model validation. The comparisons between the model's simulations and observations are exhaustive and detailed, with very informative focus on regions of special interest. Currents data processing and inherent limitations are well presented and discussed.
Thank you

The figures where model and observed currents ellipses are very interesting, however the red colored observed ellipses are sometimes hardy distinguishable of the background currents amplitude pixels.
Yes, that is true in some cases, and is why we do not rely on the reader being able to see all the observed ellipses in every Figure. To deal with this problem, we have 1) shown the comparisons at either 2 or 3 scales: national (5000km, e.g. Fig 5), regional (500km, e.g. Fig 6) and local (100km, e.g. Fig.7), 2) chosen velocity scales carefully for each Figure to reach a compromise between overlapping ellipses in strong current regions and invisible ones where the amplitude is low, 3) listed region-averaged tabulated statistics of the model-obs comparisons both on the Figures and in the Tables, and 4) listed site-specific model-obs comparisons for all current meters. We don't think there is much more we can do without including very many local-scale Figures.

I might suggest showing the model grid itself in an additional figure.
This is what Figures 1 and 2 are.

Same remark about tidal heights vector errors in addition to the modelled/observed amplitude and phase superimposed ones.
Tidal heights are not the principal focus of this paper (as is made clear in several places, starting with the title of the paper). Nevertheless, we have included model-obs height comparisons for completeness. Fig 3 and 4 show the model-obs comparisons for amplitude and phase separately, which we think is more illuminating than showing just the vector error (the combination of both components of the error). Table 2 lists statistics of amplitude, phase and also the combined (vector) error, averaged over regions. There are too many sites to include a heights-equivalent to Table 3.

The model is based on a new implementation of shallow water dynamics on an unstructured grid. As far as I understood, COMPAS model is a local evolution of the MPAS one, or at least inspired from it. Unlike the work made on the tidal observation compilation and processing, I find the modelling work rather not sufficiently convincing.
We are sorry to hear that, and have tried to make it more convincing, without repeating too much material from Herzfeld (2020) that documented the details of the model. We have emphasized to readers (at the beginning of the Model configuration section, see below) that this paper focusses on our assessment of the model, not its construction.

My first remarks concern the model grid design and setting. COMPAS developers made the choice of a basically hexagonal grid (and subsequent finite volume discretization). Despite some flexibility to tune the model resolution, it is much less flexible than triangle element grids, especially in following precisely the coastal geometry. Authors may comment on their choice.

Sorry, we disagree. COMPAS uses the dual of a Delaunay triangulation (a Voronoi diagram). Compared to using triangles, this is less prone to spurious short wave generation on a C grid. It can boundary-fit coastlines to the same degree as triangles (rays toward infinity in the Voronoi dual are truncated to the coast). COMPAS and MPAS-O are quite different in the way the coastline geometry/discretisation is treated, with COMPAS able to conform to the shoreline directly, while MPAS-O cannot. An example of a COMPAS coastline-fitted mesh is included below.

[Figure]

We have added text stating that certain aspects of COMPAS differ to MPAS in that they are coastally optimized.

The model resolution constraints (depth and currents magnitude) are also a bit surprising to me. In tidal applications, coastal geometry complexity, tidal wavelength (theoretically related to square root of depth, but possibly strongly controlled by local coastal geometry/dynamical resonance) and depth's slope related tidal currents variability scales are the most efficient constraints in setting the appropriate local resolution, especially when tidal currents are specifically targeted. I'd like authors to comment on that.

We do indeed use the sqrt(gH) wavelength in setting resolution. We also add higher resolution as distance-to-coast, so again, more agreement in terms of 'coastal geometry'. We've used the magnitude of tidal currents, rather than grad(H) as an additional refinement metric, to give more detail in the high-speed areas of particular interest. We have clarified this in the manuscript.

The setting of bathymetry is mostly set from the best available global datasets for Australian Waters, still I wonder about the choice to extend the uncovered areas with DBDB2, which is a rather ancient bathymetry database. Authors may comment on their choice.

We agree that bathymetry choice is vital to improving performance, and is a priority for future model development. To this end, we hope to capitalise on the results of the ausSeabed initiative (http://www.ausseabed.gov.au/about). We have emphasized this more in the manuscript.

The setting of the minimum model depth suggests to me that wetting/drying capabilities were not available/used in the tidal simulations. This is by itself an annoying limitation, but also minimum depth settings can significantly change the model results and, in case where the original bathymetry dataset is accurate enough, deteriorate the simulation accuracy (reversely, a 5 to 10 m minimum depth setting can help to partly compensate for bathymetry inaccuracy in nearshore regions). I'd like authors to comment on that.

We now say: "COMPAS can be run with wetting and drying activated, not only for entire water columns, but also for individual layers as sea level falls or rises. For the present application, however, neither of these capabilities were exercised to any degree; the latter because the model was run in 2D mode. Lacking adequate near-shore bathymetry for much of this large country, we chose not to attempt to properly model the tides in the inter-tidal zone, and set the minimum depth (at zero tide) to 4 m for most of the grid, but 8 m where the tides are large in the NW, NE and in Gulf St Vincent. A channel of 12 m was manually included in King Sound (in the NW) to correct an obvious error there. A similar bathymetry correction was also made in Western Port (near Melbourne). These two manual corrections had significant effect on the local tidal response, and it is anticipated that further model improvement will follow from corrections throughout the domain based on a more complete set of observations of the real topography."

My second set of remarks concerns the tidal forcing and dissipation. First having the best performances with the tidal potential left off is not a good indicator of the model performances. Also tidal loading and self-attraction forcing terms are not mentioned at all, I guess they are just no considered in COMPAS. If I am right, this is a very annoying omission for accurate tidal modelling.

Simulations were trialled with tidal potential included (equilibrium tide + self-loading/attraction). Results were found not to differ significantly from when they were absent. There is a cost to including these terms, as computation of the right ascension of the ascending node for the moon is expensive when computed at every grid point. Any changes to the solution did not warrant this additional expense. It appears that when the ratio of open boundary length (where the tide is imposed) to surface area is large, the effect of tidal potential on the solution is diminished, with the major contributor to forcing being the boundary forcing.

We have added that self-attraction/loading was trialled, and a reference to the tidal potential method used. "Tidal potential forcing and tidal self-attraction/loading (using the method of Sakamoto et al., 2013) is optionally applied in the model but we found that it made very little difference (excepting the run time) compared with other parameters such as friction, so we have omitted it for the long (1 year) run of the model described here."

Equally important, the barotropic tides generate internal tides when their energy fluxes propagate across the shelf slope, and then are partly dissipated by the subsequent barotropic to baroclinic energy conversion. This is a quite large contributor to the barotropic tides dissipation, and it must be implemented through a parameterization in depth-averaged tidal models to reach the best accuracy, even at regional scales. Again, this point is not mentioned in the paper, I just can guess that such a convenient parameterization is not available in COMPAS.

The model was run in barotropic mode only. Baroclinic energy conversion is currently not available in 2D COMPAS simulations. We have now mentioned in the manuscript that a 3D baroclinic version is under development, which would address these issues explicitly: "In this paper, we assess the ability of this model to simulate barotropic tides (both currents and sea level) as a first step towards a baroclinic model of the tides, and then a baroclinic model with non-tidal flows as well."

Many places in the Australian Waters are very challenging in terms of tidal dynamics, and will require raising the COMPAS tidal capabilities to a more comprehensive level, or at least discuss the impact of the missing tidal ingredients. I'd like authors to comment on these critical issues.

We certainly agree that our diverse tidal environment provides a significant challenge, especially since the bathymetry is uncertain in places, and there are inevitably some errors remaining in both the parent model and the validation data set. The importance of baroclinic processes can not be denied either. The paper now has a new final sentence: We conclude by reminding readers that the work reported here is just an initial step towards a more complete description of Australia's tides, which will potentially include 1) the variation in the vertical dimension of the tidal currents, 2) finer horizontal resolution, 3) more accurate sea-floor topography, 4) more accurate offshore boundary conditions, and 5) within-domain tidal potential forcing and self-attraction.

Last but not least, the open boundary conditions setting can be potentially critical in the overall simulations accuracy, their discussion in section 2 could be complemented with a domain-wide vector difference between the forcing atlas (TPXO) and COMPAS results.
Thank you for the suggestion but we think this comparison with another model (the one we are nesting inside), while interesting to some readers, would be a distraction from the main emphasis of the paper, which is the assessment of our model against observations. There is also the question of which version of TPXO should we compare to? The one we nest inside (1/6°) or 1 or more versions of the 1/30° 'Atlas' product? We looked at this and decided to make just a short sentence summarising the salient facts (see the paragraph at end of section 7, now slightly edited to remind readers that tidal potential forcing is inactive in the present version of our model)

In summary, the observational and comparison sections are very informative and well organized, and I think they are fully suited for publication. Reversely, the modelling part really needs to be augmented/revised/strengthened. Consequently, I encourage the authors to make the necessary changes to the modeling sections to reach the same level of scientific value as for the observational ones. In consequence, I will consider publication after a major revision of the modeling discussion, with no doubt that the authors will be successful in submitting a more appropriate version. I will be happy to review any new submission, and will provide a more detailed review at this occasion as the present version is susceptible to significantly vary in the revised one.
We have made some small augmentations of the modelling section this paper but, as mentioned above, we have avoided repeating too much material from Herzfeld et al (2020) which documents the details of the model. The present paper focusses on our assessment of the model, not its construction. To clarify the scope of the paper, we have added the following text at the beginning of the Model configuration section:
As mentioned above, the work reported here was done for two reasons 1) to identify regions where tidal currents are prospective from a renewable energy point of view, 2) to lay the foundations of a more general-purpose national model of the tidal currents of Australia. The model we used is called COMPAS (Coastal Ocean Marine Prediction Across Scales). It is a fully non-linear 3D model that has been described in full by Herzfeld et al., (2020). In this paper, we assess the ability of this model to simulate barotropic tides (both currents and sea level) as a first step towards a baroclinic model of the tides, and then a baroclinic model with non-tidal flows as well.

**Community Comment (Roger Proctor)**

This paper describes the results of tidal simulations using a new unstructured grid model for Australian coastal waters, initially developed for a tidal renewable energy project. The model results, from depth-averaged simulations, are compared with observations from an unprecedented collection of tidal height and tidal current locations at which a minimum of 11 tidal constituents are available. This assembly of observed tidal constituents is valuable in its own right, and the published model tidal constituents form a useful dataset. The paper is divided into sections describing the

model setup and preliminary experiments, the two observational datasets, the model-observation analysis methodology, followed by the results and a discussion. A comprehensive set of statistics is offered, resulting in a regional approach to assessing the quality of the model results. Overall the paper offers the reader several new perspectives: on the observation coverage of the tides around Australia; on the diversity of its tidal regimes; and on the ability of this new model to accurately represent these regimes. As such it is a valuable contribution to the journal and the published datasets of value to the community.

Thank you for the kind words

Some thoughts and suggested minor modifications are discussed below.

The discussion of model configuration suggests the use of the unstructured grid is a computational saving, indicating a regular grid model of similar resolution would require 1.5 million points to match the 'mean resolution' (not defined). This is not a large array for a simple 2D model so the saving, if any, may not be great. The smallest cell in the unstructured mesh is ~330m which is relatively large for some of the areas in question. I wondered if the computational constraints of the explicit scheme was limiting the calculation.

A model using 1.5 million surface cells is tractable, however, this will always run slower than one using just 12% as many cells, all other things being equal. Given that over 70 simulations were performed during the optimization procedure using a very modest number of processors, this saving in wall-time or CPU cost is non-negligible.

Although certain regions of the model are likely under-resolved, we considered this first attempt at a national model a good balance between accurately capturing the broad tidal circulation patterns and model throughput. We have added text to this effect in the manuscript, and also added the mean distance between centres (2100 m).

Since the simulations were conducted in 2D mode, semi-implicit approaches (essentially an implicit model in 2D mode) would be expected to increase throughput due to increased timesteps. However, the semi-implicit approach does have its drawbacks, notably, it is difficult to modularize open boundary conditions that can be 'mixed and matched', due to the explicit coding of these schemes as source terms into the matrix inversion procedure. Such models typically have quite a limited array of open boundary conditions, which may hinder optimization of the open boundary problem.

Lines 75-80 discuss the bathymetry used, and points to use of minimum depths, which would limit any wetting and drying, which may impact on results with large tidal range; was this tested in the preliminary experiments?

Yes – see discussion above.

Line 90+ describes the open boundary set up which is indeed quite unusual. A sentence or two to explain why this works would be helpful, particularly on how internally generated motions reaching the open boundary are handled.

Agreed, we now say: This situation is quite unusual, and suggests that the TPXO values at the boundary are largely in tune with the interior dynamics of the model (even though TPXO and COMPAS have their differences), obviating the need for strategies to make the boundary transmissive to outgoing signals.

Line 100+ describes the intitial experiments conducted to arrive at the finally chosen parameter settings (e.g. drag coefficient). Given that later in the paper, in discussing the results, there are several assertions as to discrepancies between model and observation, e.g. line 375, line 388, could these initial experiments offer any explanations?

There were 72 simulations performed during the optimization process. There were some step changes towards convergence to a skilful solution. Using TPXO on its native grid was one such step. These optimizations have led us to believe that friction modifications have negligible impact, tidal body force has a very small impact, open boundary configuration and bathymetry changes has a large impact. The open boundaries are now well optimized, and it is expected that further bathymetry improvements would decrease model-data discrepancies. We have emphasized the need for improved bathymetry in the manuscript.

Line 135-140 … how close to the island? The text seems to suggest that the model cell size may also need refining to capture the variability.
Table 3 lists all instrument positions. Distances to reefs on the GBR may be as low as 1km but this is uncertain due to bathymetry errors (see the differences between modelled and recorded depths) so we chose not to try and define 'close'. The point is that islands or reefs are close enough to matter.

Line 155 … 'for all the usual reasons' might need an explanation.
We have added: ", some of which are 1) the nature of model (and observation) errors is likely to differ significantly depending on the constituent frequency and amplitude, 2) errors of the ellipse orientation are then easily distinguished from errors of the phase and major axis length, all of which impact differently on various users, 3) it is the most succinct way of describing the data set."

Line 174, the penalty function; this is dimensionally imbalanced and needs an explanation for the D/5C component.
Sorry, this was not clear, as discussed above

Many of the figures, e.g. Figure 3, include tables of percentiles. Provide a sentence explaining these.
Sorry, we thought the caption to Figure 3 was sufficient. We have added ('%') to explain the use of that symbol.

Similarly, some tables (e.g. Table 2), have '%obs' values which need an explanation.
We've added: The %obs row expresses the RSS values in the line above as a percentage of the observed RSS.

Line 281 refers to sites in Banks Strait but in the table they are labelled Bass.
Table 3 is now fixed, thank you.

Line 356, spell out RIB.
We've changed this to 'speedboats'

Line 380 … it would be helpful to have Broad Sound marked on Figure 11.
Done, see above

Line 384 … explain why you query the mechanical current meters.
Did you overlook the next sentence, or want it expanded on? "Due to limited storage capacity, the flow direction was only sampled instantaneously once an hour, so short-period changes of direction were not averaged." We've now added "To minimise noise due to waves (i.e, rectified orbital velocities spinning the rotor even when the current velocity is zero - Griffin, 1988)
Griffin (1988): Mooring Design to minimize Savonius rotor overspeeding due to wave action.

Mark Lady Musgrave on a figure.
Done, see above

Line 394 … suggest changing 'the amplitude of S2 exceeds that of M2 (barely),' to say 'the amplitude of S2 is of similar magnitude to that of M2,'.
Good idea. We've changed it to "the amplitudes of S2 and M2 are nearly the same,"

Line 408 … 'and thus underestimates the errors'. How do you know?
Fair point. Neglecting the internal tide does little damage to the depth-mean velocity. We were thinking of users who will use our prediction of the depth-mean as a prediction of the tide at all depths. We have removed "and thus underestimates the errors".

Line 415 … Given that the official predictions are available, might be a useful addition if you did compare. Even to demonstrate the adequacy, or otherwise, of the official predictions.
We will propose this to BoM (who issue the official predictions).

Line 416+ This doesn't offer an explanation of why the you think the tidal currents are poorly predicted in this region.
That is because 1) we are not sure of the reason, but have now added "It appears that this problem is largely inherited from the boundary conditions", 2) it is a low-priority mystery, for the reason given (tidal currents are very small compared to non-tidal).

Line 430 … As we know, M4 and other higher harmonics are generated internally through non-linear model terms. Do you have anything to say on this generation mechanism within the model?
Lacking any evidence that the mechanism in the model is faithful to the real world, we'd rather not speculate on this. We've reworded this: where amplitudes up to 5.9 cm s$^{-1}$ were observed (Fig. 13). Model amplitudes are comparable (up to 4.3 cm s$^{-1}$) but there is not much correspondence with the observations. Given the complexity of both the observed and the modelled currents, and relatively small contribution to the total, we can't be confident that the modelled M4 velocities are accurate enough to warrant inclusion of these constituents when making predictions.

Line 441 … Can I suggest rewriting this sentence 'Over the continental shelf, this is the case for the southern half of the continent from Ningaloo Reef in the west to Fraser Island in the east, excepting Bass Strait and the South Australian gulfs (i.e. the sections where the shelf is narrow).' as " Over the continental shelf, this is the case for the southern half of the continent from Ningaloo Reef in the west to Fraser Island in the east (i.e. the sections where the shelf is narrow).' Exceptions are Bass Strait and the South Australian gulfs."
Hmm, we're not sure that's any better. So we've removed the bit in brackets, leaving "Over the continental shelf, this is the case for the southern half of the continent from Ningaloo Reef in the west to Fraser Island in the east, excepting Bass Strait and the South Australian gulfs."

Line 480 … Whilst the focus of the paper is on tidal currents, the statement that non-tidal currents play an important role in many parts of the Australian coastal domain leads the reader to wonder whether future versions of the model will attempt to provide this missing component. In this context, lessons learnt by Witeranje et al (2018) may be useful. Also, some insight into what improvements are intended (or are in development) and why these are seen as improvements would be useful.
Non-tidal currents, as you know, is a totally different modelling problem, and not one that we want to discuss in this paper.

Ref: Wijeratne, S., Pattiaratchi, C., & Proctor, R. (2018). Estimates of surface and subsurface boundary current transport around Australia. Journal of Geophysical Research: Oceans, 123, 3444–3466. https://doi.org/10.1029/2017JC013221

---

## Author Response (AR2)

Dear Dr Valcke,

Thank you for your comments on our resubmitted manuscript. We have taken all your comments into account and have revised the paper as follows:

*1) Preliminary tests and conclusions*

*In your reply, you write: "There were 72 simulations performed during the optimization process. There were some step changes towards convergence to a skilful solution. Using TPXO on its native grid was one such step. These optimizations have led us to believe that friction modifications have negligible impact, tidal body force has a very small impact, open boundary configuration and bathymetry changes has a large impact."*

*Is this clearly stated in the manuscript? If so, can you point me to the exact lines? If not, can you add some text detailing those conclusions?*

Lines 105-111 state the surprising result regarding the boundary conditions and no apparent need for flux adjustments (as a radiation condition).

The adjustable model parameters are now listed in lines 122-131 in a new way, as follows:

The model parameters (with final values in brackets) adjusted during the series of test runs included: 1) the bottom drag coefficient (0.003), 2) spatial variations of bottom drag (off), 3) bottom drag scheme (quadratic), 4) coastal depth (4 or 8m, see above), 5) horizontal viscosity ($350-2 \times 10^5 m^2 s^{-1}$, scaling with cell size), 6) bathymetry filtering (median), 7) flux adjustment timescale (not applicable, see above), 8) tidal potential forcing and tidal self-attraction/loading (using the method of Sakamoto et al., 2013) (off), 9) bathymetry data source (see above) and 10) interior relaxation to TPXO (off). These experiments proceeded in an ad-hoc search for closer agreement with the observations. Apart from this 'model tuning', no data assimilation was used with these model runs. Perhaps the most surprising results of these tests were that 1) flux adjustment was not needed (see above), 2) tidal potential forcing and self-attraction/loading did not reduce model error significantly (while nearly doubling the model cost), and 3) model errors were not overly sensitive to bottom friction.

*Similarly, in your reply, you write that the impact of using minimum depths, which limits the wetting and drying, was tested in preliminary experiments, but I don't see where this would be stated (at least not in the 3rd paragraph pn p.5 where wetting and drying is discussed).*

You are correct, p5 is the only place this is mentioned, apart from where it appears briefly in lines 122-131 copied above.

*2) Figure 3*

*Please make a clearer relation between the captions and the columns appearing in the insert; I suppose that "1) the whole model height field " refers to the column "model", that "2) m=model at validation sites" refers to the column "@obs", that "3) model error m-o" refers to the column "model-obs"; furthermore I think the column "615 obs" is not described in the captions.*

This caption is now improved, thank you.

**Figure 1 M2 height amplitude as a colour-fill map (the model) and points (observations), and inset as a quantity-quantity plot. Statistics listed are percentiles ('%') of 1) at left, the model height field at all grid points, 2) the model at observation sites (hereafter m), 3) model error m-o and 4) the observed values o (of which there are 615 within the area shown). At right are 1) <|m-o|>, the mean of the absolute value of model error m-o, 2) <m-o>, the mean error, and 3) <m> and <o> which are the means of m and o. A log scale is used, starting at 10cm, so not all points can be shown.**

*3) Root Sum Square*

*Can you define the Root Sum Square when you introduce it on p.9 and harmonize the wording? I.e. you write either "Root Sum Square" as in the abstract, "Root Sum of Squared" as on p.9, "root sum of squares" as in Table 2 and Table 4, "root sum squared" as on p.36 .*

We have now inserted a displayed equation, and used consistent wording, thank you.

*4) Model grid*

*As asked by reviewer #2, can you add a figure with the model grid ? Fig.1 shows only the value of the grid spacing and Fig.2 the model depth.*

As discussed by email, we don't think it is possible to draw a map showing polygons down to 1km or less when the map covers the whole of Australia.

*5) p.5, l.85: change dbdb2 for DBDB2*

Done, thank you.

*6) p.31, l 410-411: I think "there" should be removed in "since the modelled M2 amplitude there is nearly (within about 10%) as great as the observed value. "*

Done, thank you.

---

## Author Response (AR3)

Dear Dr Valcke,
Thank you for your comments on our resubmitted manuscript.

Thank you for this revised version of the manuscript. I am a bit sorry to say that I consider you did not properly answered two of my comments, so I still ask you to consider the following modifications in the next version of your manuscript.

First, when I wrote "Similarly, in your reply, you write that the impact of using minimum depths, which limits the wetting and drying, was tested in preliminary experiments, but I don't see where this would be stated" you simply answered that "You are correct, p5 is the only place this is mentioned, apart from where it appears briefly in lines 122-131 copied above." This is not what I was expecting! First, contrary to what you answered, I don't see any justification of using minimum depths on p.5. What I am asking is that you clearly state in the manuscript that the impact of using minimum depths, which limits the wetting and drying, was tested in preliminary experiments and that the conclusion of those preliminary experiments justified using minimum depths. Please add something about this in the manuscript.

Second, I think the Figure 3 captions still need clarification. What I am asking is a clear correspondence between the text in the captions and the different columns. So please consider modifying (again) the 2nd sentence of the captions as follows:
"Statistics listed are percentiles ('%') of 1) the model height field at all grid points (column "model" at left), 2) the model at observation sites, hereafter "m" (column "@obs", 3) model error (column "model-obs") and 4) the observed values o, of which there are 615 within the area shown (column "615 obs").

Of course, let me know if you don't agree with or don't understand those remarks ...

We think both comments are totally valid and have revised the paper as follows:

**Minimum depths, wetting and drying.**
We have substantially rewritten the paragraph on page 5 where wetting and drying is mentioned, to make it clearer what we did, why we used minimum depths, and what the consequence was. It now reads:

COMPAS can be run with wetting and drying activated, not only for entire water columns, but also for individual layers (in a 3D application) as sea level falls or rises. For the present (2D) application, wetting and drying was not activated other than in preliminary test runs. The main problem with having wetting and drying activated was that it made comparison with tide gauges difficult. At many tide gauge sites, the model cells near the gauge dried at low tide but the observations showed drying at the exact location of the tide gauge did not occur – presumably because the gauge is sited within a harbour or shipping channel unresolved by the model mesh. We chose to deal with this problem by preventing drying by setting the minimum depth (at zero tide) to 8 m at the coast in regions where the tides are large (impacting cells totalling 0.6% of the total model area, mostly in the southern GBR or the region around Darwin) or 4m elsewhere (impacting cells totalling 1.4% of the model area, mostly in the Gulf of Carpentaria). The impact of this workaround solution on the nature of the tides, outside the impacted cells, was evidently negligible. A channel of 12 m was manually included in King Sound (in the NW) to correct an obvious error there, greatly improving the accuracy of the model in this location where Australia's greatest tides are to be found. A similar manual bathymetry correction was also made in Western Port (near Melbourne). We anticipate that further local improvements will follow from the use of an even finer mesh and a more complete set of observations of the real topography.

**Figure 3 caption**

This now reads (following your suggestion very closely):

M2 height amplitude as a colour-fill map (the model) and points (observations), and inset as a quantity-quantity plot. Statistics listed are percentiles ('%' columns) of 1) the model height field at all grid points ('model' column at left), 2) the model at observation sites, hereafter 'm' ('@ obs' column), 3) model error ('model-obs' column), and 4) the observed values 'o' of which there are 615 within the area shown ('615 obs' column). At far right are listed $<|m-o|>$, the mean of the absolute value of m-o, $<m-o>$, the mean error, or bias, and $<m>$ and $<o>$, the mean modelled and observed amplitudes. A log scale is used, starting at 10cm, so not all points can be shown.

We also made minor edits to the headings of the Tables 2 and 4 to make them more compatible, and to clarify the units of each sub-table, e.g. (MAE, cm)

Thank you very much for all your work. We hope you find this version satisfactory for the Journal.

---

## Author Response (AR4)

Dear Dr Valcke,

Thank you for accepting our paper. We will be very pleased to see it published in GMD.

Dear Editorial staff,

1) You might like to investigate why I did not receive an email advising me that the paper was accepted. I only found out when one of my co-authors said they had got an email from you. I checked my email carefully, including Junk folder.

2) Dr Valcke did not ask for any more changes but I have made one minor change in the Data Available section, tidying up the reference to the Herzfeld et al data collection.

3) I have made 2 zip archives of all the figures. The pdf files are much nicer but 3 of them exceed your 5Mb limit. I cannot see how to reduce their size. Figures 1 and 2 are less important than the rest, so please use the png versions of these if you want to keep the total size down.

Best regards

David